# Fast Distance Oracles for Any Symmetric Norm

**Yichuan Deng**
University of Science and Technology of China
ethandeng02@gmail.com

**Zhao Song**
Adobe Research
zsong@adobe.com

**Omri Weinstein**
The Hebrew University and Columbia University
ow2161@columbia.edu

**Ruizhe Zhang**
The University of Texas at Austin
ruizhe@utexas.edu

## Abstract

In the *Distance Oracle* problem, the goal is to preprocess $n$ vectors $x_1, x_2, \ldots, x_n$ in a $d$-dimensional metric space $(\mathbb{X}^d, \| \cdot \|_l)$ into a cheap data structure, so that given a query vector $q \in \mathbb{X}^d$ and a subset $S \subseteq [n]$ of the input data points, all distances $\|q - x_i\|_l$ for $x_i \in S$ can be quickly approximated (faster than the trivial $\sim d|S|$ query time). This primitive is a basic subroutine in machine learning, data mining and similarity search applications. In the case of $\ell_p$ norms, the problem is well understood, and optimal data structures are known for most values of $p$.

Our main contribution is a fast $(1 \pm \varepsilon)$ distance oracle for *any symmetric* norm $\| \cdot \|_l$. This class includes $\ell_p$ norms and Orlicz norms as special cases, as well as other norms used in practice, e.g. top-$k$ norms, max-mixture and sum-mixture of $\ell_p$ norms, small-support norms and the box-norm. We propose a novel data structure with $\widetilde{O}(n(d + \mathrm{mmc}(l)^2))$ preprocessing time and space, and $t_q = \widetilde{O}(d + |S| \cdot \mathrm{mmc}(l)^2)$ query time, for computing distances to a subset $S$ of data points, where $\mathrm{mmc}(l)$ is a complexity-measure (concentration modulus) of the symmetric norm. When $l = \ell_p$, this runtime matches the aforementioned state-of-art oracles.

## 1 Introduction

Estimating and detecting similarities in datasets is a fundamental problem in computer science, and a basic subroutine in most industry-scale ML applications, from optimization [CMF+20, CLP+20, XSS21] and reinforcement learning [SSX21], to discrepancy theory [SXZ22] and co-variance estimation [Val12, Alm19], Kernel SVMs [CS09, SSSSC11], compression and clustering [IRW17, MMR22], to mention a few. Such applications often need to quickly compute distances of online (test) points to a subset of points in the input data set (e.g., the training data) for transfer-learning and classification. These applications have motivated the notion of *distance oracles* (DO) [Pel00, GPPR04, WP11]: In this problem, the goal is to preprocess a dataset of $n$ input points $X = (x_1, x_2, \ldots, x_n)$ in some $d$-dimensional metric space, into a small-space data structure which, given a query vector $q$ and a subset $\mathsf{S} \subseteq [n]$, can quickly estimate all the distances $\mathsf{d}(q, x_i)$ of $q$ to $\mathsf{S}$ (note that the problem of estimating a *single* distance $\mathsf{d}(q, x_i)$ is not interesting in $\mathbb{R}^d$, as this can be trivially done in $O(d)$ time, which is necessary to merely read the query $q$). The most well-studied case (in both theory and practice) is when the metric space is in fact a *normed* space, i.e., the data points $\{x_i\}_{i \in [n]} \in \mathbb{R}^d$ are endowed with some predefined norm $\| \cdot \|$, and the goal is to quickly estimate $\|x_i - q\|$ simultaneously for all $i$, i.e., in time $\ll nd$ which is the trivial query time. Distance oracles can therefore be viewed as generalizing *matrix-vector* multiplication: for the inner-product distance function $\langle x_i, q \rangle$, the query asks to approximate $X \cdot q$ in $\ll nd$ time.

36th Conference on Neural Information Processing Systems (NeurIPS 2022).

For the most popular distance metrics—the Euclidean distance ($\ell_2$-norm) and Manhattan distance ($\ell_1$-norm)—classic dimension-reduction (sketching) provide very efficient distance oracles [JL84, AC09, LDFU13]. However, in many real-world problems, these metrics do not adequately capture similarities between data points, and a long line of work has demonstrated that more complex (possibly *learnable*) metrics can lead to substantially better prediction and data compression [DKJ$^+$07]. In particular, many works over the last decade have been dedicated to extending various optimization problems beyond Euclidean/Manhattan distances, for example in (kernel) linear regression [SWY$^+$19], approximate nearest neighbor [ANN$^+$17, ANN$^+$18], sampling [LSV18], matrix column subset selection [SWZ19], and statistical queries [LNRW19].

For $\ell_p$ norms, the DO problem is well-understood [BYJKS04], where the standard tool for constructing the data structure is via randomized *linear sketching* : The basic idea is to reduce the dimension ($d$) of the data points by applying some *sketching matrix* $\Phi \in \mathbb{R}^{m \times d}$ ($m \ll d$) to each data point $x_i$ and store the sketch $\Phi x_i \in \mathbb{R}^m$. For a query point $q$, linearity then allows to estimate the distance from $\Phi(q - x_i)$. The seminal works of [JL84, AMS99, CCF04, TZ12] developed polylogarithmic-size sketching methods for the $\ell_2$-norm, which was extended, in a long line of work, to any $\ell_p$ norm with $0 < p < 2$ [Ind06, KNPW11, CN20]. For $p > 2$, the sketch-size ($d$) becomes *polynomial*, yet still sublinear in $d$ [SS02, BYJKS04].

In this work, we consider general *symmetric norms*, which generalize $\ell_p$ norms. More formally, a norm $l : \mathbb{R}^d \to \mathbb{R}$ is symmetric if, for all $x \in \mathbb{R}^d$ and every $d \times d$ permutation matrix $P$, it satisfies $l(Px) = l(x)$ and also $l(|x|) = l(x)$, where $|x|$ is the coordinate-wise absolute value of $x$ (for a broader introduction, see [Bha97] Chapter IV). Important special cases of symmetric norms are $\ell_p$ norms and *Orlicz norms* [ALS$^+$18], which naturally arise in harmonic analysis and model data with sub-gaussian properties. Other practical examples of symmetric norms include top-$k$ norms, max-mix of $\ell_p$ norms, sum-mix of $\ell_p$ norms, the $k$-support norm [AFS12] and the box-norm [MPS14].

Several recent works have studied dimension-reduction (sketching) for special cases of symmetric norms such as the Orlicz norm [BBC$^+$17, SWY$^+$19, ALS$^+$18], for various numerical linear algebra primitives [SWZ19]. These sketching techniques are quite ad-hoc and are carefully tailored to the norm in question. It is therefore natural to ask whether *symmetry alone* is enough to guarantee dimensionality-reduction for symmetric similarity search, in other words:

*Is there an efficient $(1 + \epsilon)$-distance oracle for general symmetric norms?*

By "efficient", we mean small space and preprocessing time (ideally $\sim nd$), fast query time ($\ll d|S|$ for a query $(q, S \subseteq [n])$) and ideally supporting dynamic updates to $x_i$'s in $\widetilde{O}(d)$ time. Indeed, most ML applications involve rapidly-changing *dynamic datasets*, and it is becoming increasingly clear that static data structures do not adequately capture the requirements of real-world applications [JKDG08, CMF$^+$20, CLP$^+$20]. As such, it is desirable to design a *dynamic distance oracle* which has both small update time ($t_u$) for adding/removing a point $x_i \in \mathbb{R}^d$, and small query time ($t_q$) for distance estimation. We remark that most known DOs are dynamic by nature (as they rely on linear sketching techniques), but for general metrics (e.g., graph shortest-path or the $\ell_\infty$ norm) this is much less obvious, and indeed *linearity* of encoding/decoding will be a key challenge in our data structure (see next section). The problem is formally defined as follows:

**Definition 1.1** (Symmetric-Norm Distance Oracles). *Let $\| \cdot \|_{\mathrm{sym}}$ be the symmetric norm. The* Symmetric-norm Distance Oracle *is a data structure that efficiently supports any sequence of the following operations:*

- INIT($\{x_1, x_2, \cdots, x_n\} \subset \mathbb{R}^d, \epsilon \in (0, 1), \delta \in (0, 1)$). *The data structure takes $n$ data points $\{x_1, x_2, \ldots, x_n\} \subset \mathbb{R}^d$, an accuracy parameter $\epsilon$ and a failure probability $\delta$ as input.*

- UPDATEX($z \in \mathbb{R}^d, i \in [n]$). *Update the data structure with the $i$-th new data point $z$.*

- ESTPAIR($i, j \in [n]$) *Outputs a number* pair *such that $(1 - \epsilon)\|x_i - x_j\|_{\mathrm{sym}} \leq$ pair $\leq (1 + \epsilon) \cdot \|x_i - x_j\|_{\mathrm{sym}}$ with probability at least $1 - \delta$.*

- QUERY($q \in \mathbb{R}^d$). *Outputs a vector* dst $\in \mathbb{R}^n$ *such that $\forall i \in [n], (1 - \epsilon)\|q - x_i\|_{\mathrm{sym}} \leq$ dst$_i \leq (1 + \epsilon)\|q - x_i\|_{\mathrm{sym}}$. with probability at least $1 - \delta$.*

*where $\|x\|_{\mathrm{sym}}$ is the symmetric norm of the vector $x$.*

This problem can be viewed as an *online* version of the (approximate) *closest-pair* problem [Val12], which asks to find the closest pair of points among an *offline batch* of data points $X = x_1, \ldots, x_n \in \mathbb{R}^d$, or equivalently, the smallest entry of the *covariance matrix* $XX^\top$. One major (theoretical) advantage of the offline case is that it enables the use of fast matrix-multiplication (FMM) to speed-up the computation of the covariance matrix [Val12, AWY14, AWY18, Alm19] (i.e., sub-linear amortized per query). By contrast, in the online setting such speedups are conjectured to be impossible [HKNS15, LW17].

**Notations.** For any positive integer $n$, we use $[n]$ to denote $\{1, 2, \ldots, n\}$. For any function $f$, we use $\widetilde{O}(f)$ to denote $f \cdot \text{poly}(\log f)$. We use $\Pr[\cdot]$ to denote probability. We use $\mathbb{E}[\cdot]$ to denote expectation. We use both $l(\cdot)$ and $\|\cdot\|_{\text{sym}}$ to denote the symmetric norm. We use $\|\cdot\|_2$ to denote the entry-wise $\ell_2$ norm. We define a tail notation which is very standard in sparse recover/compressed sensing literature. For any given vector $x \in \mathbb{R}^d$ and an integer $k$, we use $x_{\overline{[k]}}$ or $x_{\text{tail}(k)}$ to denote the vector that without (zeroing out) top-$k$ largest entries (in absolute). For a vector $x$, we use $x^\top$ to denote the transpose of $x$. For a matrix $A$, we use $A^\top$ to denote the transpose of $A$. We use $\mathbf{1}_n$ denote a length-$n$ vector where every entry 1. We use $\mathbf{0}_n$ denote a length-$n$ vector where every entry is 0.

## 1.1 Our Results

Two important complexity measures of (symmetric) norms, which capture their "intrinsic dimensionality", are the *concentration modulus* (mc) and *maximum modulus* (mmc) parameters. We now define these quantities along the lines of [BBC$^+$17, SWY$^+$19].

**Definition 1.2** (Modulus of concentration (mc))**.** *Let $X \in \mathbb{R}^n$ be uniformly distributed on $S^{n-1}$, the $\ell_2$ unit sphere. The median of a symmetric norm $l$ is the (unique) value $M_l$ such that $\Pr[l(X) \geq M_l] \geq 1/2$ and $\Pr[l(X) \leq M_l] \geq 1/2$. Similarly, $\mathfrak{b}_l$ denotes the maximum value of $l(x)$ over $x \in S^{n-1}$. We call the ratio*

$$\text{mc}(l) := \mathfrak{b}_l / M_l$$

*the modulus of concentration of the norm $l$.*

For every $k \in [n]$, the norm $l$ induces a norm $l^{(k)}$ on $\mathbb{R}^k$ by setting

$$l^{(k)}((x_1, x_2, \ldots, x_k)) := l((x_1, x_2, \ldots, x_k, 0, \ldots, 0)),$$

where $(x_1, x_2, \ldots, x_k, 0, \ldots, 0) \in \mathbb{R}^n$.

**Definition 1.3** (Maximum modulus of concentration (mmc))**.** *Define the maximum modulus of concentration of the norm $l$ as*

$$\text{mmc}(l) := \max_{k \in [n]} \text{mc}(l^{(k)}) = \max_{k \in [n]} \frac{\mathfrak{b}_{l^{(k)}}}{M_{l^{(k)}}}$$

Next, we present a few examples (in Table 1) for different norm $l$'s $\text{mmc}(l)$.

| **Norm $l$** | $\text{mmc}(l)$ |
|---|---|
| $\ell_p (p \leq 2)$ | $\Theta(1)$ |
| $\ell_p (p > 2)$ | $\Theta(d^{1/2-1/p})$ |
| top-$k$ norms | $\widetilde{\Theta}(\sqrt{d/k})$ |
| $k$-support norms and the box-norm | $O(\log d)$ |
| max-mix and sum-mix of $\ell_1$ and $\ell_2$ | $O(1)$ |
| Orlicz norm $\|\cdot\|_G$ | $O(\sqrt{C_G \log d})$ |

Table 1: Examples of $\text{mmc}(l)$, where max-mix of $\ell_1$ and $\ell_2$ is defined as $\max\{\|x\|_2, c\|x\|_1\}$ for a real number $c$, sum-mix of $\ell_1$ and $\ell_2$ is defined as $\|x\|_2 + c\|x\|_1$ for a real number $c$, and $C_G$ of Orlicz norm is defined as the number that for all $0 < x < y$, $G(y)/G(x) \leq C_G(y/x)^2$.

We are now ready to state our main result:

**Theorem 1.4** (Main result, informal version of Theorem D.1). *Let* $\| \cdot \|_l$ *be any symmetric norm on* $\mathbb{R}^d$. *There is a data structure for the online symmetric-norm Distance Oracle problem (Definition 1.1), which uses* $n(d + \mathrm{mmc}(l)^2) \cdot \mathrm{poly}(1/\epsilon, \log(nd/\delta))$ *space, supporting the following operations:*

- INIT($\{x_1, x_2, \ldots, x_n\} \subset \mathbb{R}^d, \epsilon \in (0,1), \delta \in (0,1)$): *Given* $n$ *data points* $\{x_1, x_2, \ldots, x_n\} \subset \mathbb{R}^d$, *an accuracy parameter* $\epsilon$ *and a failure probability* $\delta$ *as input, the data structure preprocesses in time*

$$n(d + \mathrm{mmc}(l)^2) \cdot \mathrm{poly}(1/\epsilon, \log(nd/\delta)).$$

  *Note that* $\mathrm{mmc}()$ *is defined as Definition 1.3.*

- UPDATEX($z \in \mathbb{R}^d, i \in [n]$): *Given an update vector* $z \in \mathbb{R}^d$ *and index* $i \in [n]$, *the data structure receives* $z$ *and* $i$ *as inputs, and updates the* $i$-*th data point* $x_i \leftarrow z$, *in*

$$d \cdot \mathrm{poly}(1/\epsilon, \log(nd/\delta))$$

  *time.*

- QUERY($q \in \mathbb{R}^d, \mathsf{S} \subseteq [n]$): *Given a query point* $q \in \mathbb{R}^d$ *and a subset of the input points* $\mathsf{S} \subseteq [n]$, *the* QUERY *operation outputs a* $(1 + \epsilon)$- *multiplicative approximation to each distance from* $q$ *to points in* $\mathsf{S}$, *in time*

$$(d + |\mathsf{S}| \cdot \mathrm{mmc}(l)^2) \cdot \mathrm{poly}(1/\epsilon, \log(nd/\delta))$$

  *i.e. it provides a set of estimates* $\{\mathrm{dst}_i\}_{i \in \mathsf{S}}$ *such that:*

$$\forall i \in \mathsf{S}, (1 - \epsilon)\|q - x_i\|_l \leq \mathrm{dst}_i \leq (1 + \epsilon)\|q - x_i\|_l$$

  *with probability at least* $1 - \delta$.

- ESTPAIR($i, j \in [n]$) *Given indices* $i, j \in [n]$, *the* ESTPAIR *operation takes* $i$ *and* $j$ *as input and approximately estimates the symmetric norm distances from* $i$-*th to the* $j$-*th point* $x_i, x_j \in \mathbb{R}^d$ *in time* $\mathrm{mmc}(l)^2 \cdot \mathrm{poly}(1/\epsilon, \log(nd/\delta))$ *i.e., it provides an estimated distance* pair *such that:*

$$(1 - \epsilon)\|x_i - x_j\|_l \leq \mathrm{pair} \leq (1 + \epsilon)\|x_i - x_j\|_l$$

  *with probability at least* $1 - \delta$.

**Roadmap.** In Section 2, we give an overview of the techniques that we mainly use in the work. In Section 3 we give an introduction of the Sparse Recovery Data we use for sketching. In Section 4 we analyze the running time and space of our data structure with their proofs, respectively. In Section 5 we show the correctness of our data structure and give its proof. Finally in Section 6 we conclude our work.

## 2 Technique Overview

Our distance oracle follows the "sketch-and-decode" approach, which was extensively used in many other sublinear-time compressed sensing and sparse recovery problems [Pri11, HIKP12, LNNT16, NS19, SSWZ22]. The main idea is to compress the data points into smaller dimension by computing, for each data point $x_i \in \mathbb{R}^d$, a (randomized) linear sketch $\Phi \cdot x_i \in \mathbb{R}^{d'}$ with $d' \ll d$ at preprocessing time, where $\Phi x_i$ is an unbiased estimator of $\|x_i\|$. At query time, given a query point $q \in \mathbb{R}^d$, we analogously compute its sketch $\Phi q$. By *linearity* of $\Phi$, the distance between $q$ and $x_i$ (i.e., $\ell(q - x_i)$) can be trivially decoded from the sketch *difference* $\Phi q - \Phi x_i$. As we shall see, this simple virtue of linearity is less obvious to retain when dealing with general symmetric norms.

**Layer approximation** Our algorithm uses the *layer approximation* method proposed by Indyk and Woodruff in [IW05] and generalized to symmetric norms in [BBC⁺17, SWY⁺19]. Since symmetric norms are invariant under reordering of the coordinates, the main idea in [IW05] is to construct a "layer vector" as follows: for a vector $v \in \mathbb{R}^d$, round (the absolute value of) each coordinate to the nearest power $\alpha^i$ for some fixed $\alpha \in \mathbb{R}$ and $i \in \mathbb{N}$, and then sort the coordinates in an increasing order.

This ensures that the $i$-th layer contains all the coordinates $j \in [d]$ satisfying: $\alpha^{i-1} < |v_j| \leq \alpha^i$. In particular, the layer vector of $v$ has the form

$$\mathcal{L}(v) := (\underbrace{\alpha^1, \ldots, \alpha^1}_{b_1 \text{ times}}, \ \underbrace{\alpha^2, \ldots, \alpha^2}_{b_2 \text{ times}}, \ \cdots, \underbrace{\alpha^P, \ldots, \alpha^P}_{b_P \text{ times}}, \ 0, \ldots, 0) \in \mathbb{R}^d,$$

where $b_i$ is the number of coordinates in layer-$i$. More importantly, since the norm is symmetric, the layer vector $\mathcal{L}(v)$ has a succinct representation: $(b_i)_{i \in [P]}$.

Then, it suffices to estimate $b_i$ for each $i \in [P]$, where the Indyk-Woodruff sketching technique can be used to approximate the vector. At the $i$-th layer, each coordinate of $v$ is sampled with probability $P/b_i$, and then the algorithm identifies the $\ell_2$-*heavy-hitters* of the sampled vector. [BBC$^+$17] gave a criterion for identifying the important layers, whose heavy-hitter coordinates in the corresponding *sampled* vectors, is enough to recover the entire symmetric norm $\|v\|_{\text{sym}}$.

Unfortunately, this technique for norm estimation does not readily translate to estimating *distances* efficiently:

- *Too many layers:* In previous works, each data point $x_i$ is sub-sampled *independently* in $R$ layers, i.e, generates $R$ subsets of coordinates $S_i^1, \ldots, S_i^R \subset [d]$. The sketch of the query point $S(q)$ then needs to be compared to each $S(x_i)$ in every layer. Since there are $R = \Omega(n)$ layers in [BBC$^+$17] of size $\Omega(d)$ across all data points, the total time complexity will be at least $\Omega(nd)$, which is the trivial query time.

- *Non-linearity:* The aforementioned sketching algorithms [BBC$^+$17] involve nonlinear operations, and thus cannot be directly used for distance estimation.

- *Slow decoding:* The aforementioned sketches take linear time to decode the distance from the sketch, which is too slow for our application.

To overcome these challenges, we use the following ideas:

**Technique I: shared randomness**  To reduce the number of layers, we let all the data points use the same set of layers. That is, in the initialization of our algorithm, we independently sample $R$ subsets $S^1, \ldots, S^R$ with different probabilities. Then, for each data point $x_i$, we consider $(x_i)_{S^j}$ as the sub-sample for the $j$-th layer, and perform sketch on it. Hence, the number of different layer sets is reduced from $nt$ to $t$, where $t$ is the number of layers for each data point. We share the layers for all points to remove the $n$ factor. For a query point $q$, we just need to compute the sketches for $(q)_{S^1}, \ldots, (q)_{S^R}$. And the distance between $q$ and $x_i$ can be decoded from $\{\Phi(x_i)_{S^j} - \Phi(q)_{S^j}\}_{j \in [R]}$. We also prove that the shared randomness in all data points will not affect the correctness of layer approximation.

**Technique II: linearization**  We choose a different sketching method called BATCHHEAVYHITTER (see Theorem D.3 for details) to generate and maintain the *linear sketches*, which allows us to decode the distance from sketch difference.

**Technical III: locate-and-verify decoding**  We design a *locate-and-verify* style decoding method to recover distance from sketch. In our data structure, we not only store the sketch of each sub-sample vector, but also the vector itself. Then, in decoding a sketch, we can first apply the efficient sparse-recovery algorithm to identify the position of heavy-hitters. Next, we directly check the entries at those positions to verify that they are indeed "heavy" (comparing the values with some threshold), and drop the non-heavy indices. This verification step is a significant difference from the typical sparse recovery approaches, which employ complex (and time-consuming) subroutines to *reconstruct the values* of the heavy-hitter coordinates. Instead, our simple verification procedure eliminates all the *false-positive* heavy-hitters, therefore dramatically reducing the running time of the second step, which can now be performed directly by reading-off the values from the memory.

With these three techniques, we obtain our sublinear-time distance estimation algorithm. Our data structure first generate a bunch of randomly selected subsets of coordinates as the layer sets. Then, for each data point, we run the BATCHHEAVYHITTER procedure to sketch the sub-sample vector in each layer[1]. In the query phase, we call the DECODE procedure of BATCHHEAVYHITTER for the

---

[1]The total sketch size of each data point is $\text{mmc}(l)^2 \cdot \text{poly} \log d$. In the $\ell_p$-norm case with $p > 2$, $\text{mmc}(l) = d^{1/2-1/p}$ (see Table 1) and our sketch size is $\widetilde{O}(d^{1-2/p})$, matching the lower bound of $\ell_p$-sketching. When $p \in (0, 2]$, $\text{mmc}(l) = \Theta(1)$ and our sketch size is $\widetilde{O}(1)$, which is also optimal.

sketch differences between the query point $q$ and each data point $x_i$, and obtain the heavy hitters of each layer. We then select some "important layers" and use them to approximately recover the layer vector $\mathcal{L}(q - x_i)$, which gives the estimated distance $\|q - x_i\|_{\mathrm{sym}}$.

Finally, we summarize the time and space costs of our data structure. Let $\epsilon$ be the precision parameter and $\delta$ be the failure probability parameter. Our data structure achieves $\widetilde{O}(n(d + \mathrm{mmc}(l)^2))$-time for initialization , $\widetilde{O}(d)$-time per data point update, and $\widetilde{O}(d + n \cdot \mathrm{mmc}(l)^2))$-time per query. As for space cost, our data structure uses the space of $\widetilde{O}(n(d + \mathrm{mmc}(l)^2))$ in total. Note that $\mathrm{mmc}$ is defined as Defnition 1.3.

## 3 Sparse Recovery Data Structure

We design a data structure named BATCHHEAVYHITTER to generate sketches and manage them. In our design, it is a "linear sketch" data structure, and providing the following functions:

- INIT($\epsilon \in (0, 0.1), n, d$). Create a set of Random Hash functions and all the $n$ copies of sketches share the same hash functions. This step takes $\mathcal{T}_{\mathrm{init}}(\epsilon, n, d)$ time.

- ENCODE($i \in [n], z \in \mathbb{R}^d, d$). This step takes $\mathcal{T}_{\mathrm{encode}}(d)$ encodes $z$ into $i$-th sketched location and store a size $\mathcal{S}_{\mathrm{space}}$ linear sketch.

- ENCODESINGLE($i \in [n], j \in [d], z \in \mathbb{R}, d$). This step takes $\mathcal{T}_{\mathrm{encodesingle}}(d)$ updates one sparse vector $e_j z \in \mathbb{R}^d$ into $i$-th sketched location.

- SUBTRACT($i, j, l \in [n]$). Update the sketch at $i$-th location by $j$-th sketch minus $l$-th sketch.

- DECODE($i \in [n], \epsilon \in (0, 0.1), d$). This step takes $\mathcal{T}_{\mathrm{decode}}(\epsilon, d)$ such that it returns a set $L \subseteq [d]$ of size $|L| = O(\epsilon^{-2})$ containing all $\epsilon$-heavy hitters $i \in [n]$ under $\ell_p$. Here we say $i$ is an $\epsilon$-heavy hitter under $\ell_2$ if $|x_i| \geq \epsilon \cdot \|x_{\overline{[\epsilon^{-2}]}}\|_2$ where $x_{\overline{[k]}}$ denotes the vector $x$ with the largest $k$ entries (in absolute value) set to zero. Note that the number of heavy hitters never exceeds $2/\epsilon^2$.

With this data structure, we are able to generate the sketches for each point, and subtract each other with its function. And one can get the output of heavy hitters of each sketch stored in it with DECODE function. More details are deferred to Section D.2.

## 4 Running Time and Space of Our Algorithm

We first analyze the running time of different procedures of our data structure DISTANCEON-SYMMETRICNORM. See Algorithm 5, with the linear sketch technique, we spend the time of $\widetilde{O}(n(d + \mathrm{mmc}(l)^2))$ for preprocessing and generate the sketches stored in the data structure. When updating the data with Algorithm 6, we spend $\widetilde{O}(d)$ to update the sketch. And when a query comes (Algorithm 7), we spend $\widetilde{O}(d + n \cdot \mathrm{mmc}(l)^2))$ to get the output distance estimation. The lemmas of running time and their proof are shown below in this section.

**Lemma 4.1** (INIT time, informal)**.** *Given data points $\{x_1, x_2, \ldots, x_n\} \subset \mathbb{R}^d$, an accuracy parameter $\epsilon \in (0, 1)$, and a failure probability $\delta \in (0, 1)$ as input, the procedure* INIT *(Algorithm 5) runs in time*

$$O(n(d + \mathrm{mmc}(l)^2) \cdot \mathrm{poly}(1/\epsilon, \log(nd/\delta))).$$

*Proof.* The INIT time includes these parts:

- Line 12 takes $O(RLU \cdot \mathcal{T}_{\mathrm{init}}(\sqrt{\beta}, n, d))$ to initialize sketches

- Line 17 to Line 19 take $O(RUdL)$ to generate the bmap;

- Line 24 takes $O(ndRUL \cdot \mathcal{T}_{\mathrm{encodesingle}}(d))$ to generate sketches

By Theorems D.3, we have

- $\mathcal{T}_{\mathrm{init}}(\sqrt{\beta}, n, d)) = n \cdot O(\beta^{-1} \log^2 d) = O(n \cdot \mathrm{mmc}(l)^2 \log^7(d)\epsilon^{-5})$,

---

**Algorithm 1** Data structure for symmetric norm estimation: members, init, informal version of Algorithm 4 and Algorithm 5

---

1: **data structure** DISTANCEONSYMMETRICNORM                         ▷ Theorem D.1
2: **members**
3:      $\{x_i\}_{i=1}^n \in \mathbb{R}^d$
4:      $\{S_{r,l,u}\}_{r\in[R],l\in[L],u\in[U]}$                    ▷ A list of the BATCHHEAVYHITTER
5:      $\{H_{r,l,u}\}_{r\in[R],l\in[L],u\in[U]} \subset [d] \times \mathbb{R}$
6: **end members**
7:
8: **public:**
9: **procedure** INIT($\{x_1, \cdots, x_n\} \subset \mathbb{R}^d, \delta, \epsilon$)          ▷ Lemma 4.1
10:      Initialize the sparse-recovery data structure $\{S_{r,l,u}\}$
11:      Create $\{\text{bmap}_{r,l,u}\}$ shared by all $i \in [n]$     ▷ $\{\text{bmap}_{r,l,u}\}$ is list of a layer set map
12:      **for** $i \in [n], j \in [d]$ **do**
13:          **if** $\text{bmap}_{r,u,l}[j] = 1$ **then**
14:              Sample $x_{i,j}$ into each subvector $\overline{x}_{r,u,l,i}$
15:          **end if**
16:      **end for**
17:      Encode $\{x_{r,u,l,i}\}_{i\in[n]}$ into $\{S_{r,l,u}\}$
18: **end procedure**
19: **end data structure**

---

**Algorithm 2** Data structure for symmetric norm estimation: query, informal version of Algorithm 7

---

1: **data structure** DISTANCEONSYMMETRICNORM
2: **public:**
3: **procedure** QUERY($q \in \mathbb{R}^d$)                          ▷ Lemma 4.3, 5.1
4:      Encode $q$ into $\{S_{r,l,u}\}$
5:      **for** $i \in [n]$ **do**
6:          Subtract the sketch of $x_i$ and $q$, get the estimated heavy-hitters of $q - x_i$
7:          Decode the sketch and store returned estimation of heavy-hitters into $H_{r,l,u}$
8:          **for** $u \in [U]$ **do**
9:              **if** $H_{r,l,u}$ provide correct indices of heavy hitters **then**
10:                  Select it as good set and store it in $H_{r,l}$
11:              **end if**
12:          **end for**
13:          Generate estimated layer sizes $\{c_k^i\}_{k\in[P]}$
14:          $\text{dst}_i \leftarrow$ LAYERVETCORAPPROX($\alpha, c_1^i, c_2^i, \ldots, c_P^i, d$)
15:          Reset $\{H_{r,l,u}\}$ for next distance
16:      **end for**
17:      **return** $\{\text{dst}_i\}_{i\in[n]}$
18: **end procedure**
19: **end data structure**

---

- $\mathcal{T}_{\text{encodesingle}}(d) = O(\log^2(d))$.

Adding them together we got the time of

$$O(RLU\mathcal{T}_{\text{init}}(\sqrt{\beta}, n, d)) + O(RUdL) + O(ndRUL \cdot \mathcal{T}_{\text{encodesingle}}(d))$$
$$= O(RLU(\mathcal{T}_{\text{init}}(\sqrt{\beta}, n, d) + nd \cdot \mathcal{T}_{\text{encodesingle}}(d)))$$
$$= O(\epsilon^{-4}\log(1/\delta)\log^4(d) \cdot \log(d) \cdot \log(d^2/\delta \cdot \log(nd))(n \cdot \text{mmc}(l)^2 \log^7(d)\epsilon^{-5} + nd\log^2(d)))$$
$$= O(n(d + \text{mmc}(l)^2) \cdot \text{poly}(1/\epsilon, \log(nd/\delta))),$$

where the first step follows from merging the terms, the second step follows from the definition of $R, L, U, \mathcal{T}_{\text{encodesingle}}(d), \mathcal{T}_{\text{init}}$, the third step follows from merging the terms.

Thus, we complete the proof.                                                    □

**Lemma 4.2** (UPDATE time, informal). *Given a new data point $z \in \mathbb{R}^d$, and an index $i$ where it should replace the original data point $x_i \in \mathbb{R}^d$. The procedure* UPDATE *(Algorithm 6) runs in time*

$$O(d \cdot \mathrm{poly}(1/\epsilon, \log(nd/\delta))$$

*Proof.* The UPDATE operation calls BATCHHEAVYHITTER.ENCODE for $RLU$ times, so it has the time of

$$O(RLU \cdot \mathcal{T}_{\mathrm{encode}}(d)) = O(\epsilon^{-4} \log(1/\delta) \log^4(d) \cdot \log(d) \cdot \log(d^2/\delta) \cdot d \log^2(d) \cdot \log(nd))$$
$$= O(\epsilon^{-4} d \log^9(nd/\delta))$$

where the first step follows from the definition of $R, L, U, \mathcal{T}_{\mathrm{encode}}(d)$, the second step follows from

$$\log(1/\delta) \log^4(d) \log(d) \log(d^2/\delta) \log^2(d) \log(nd)$$
$$= (\log(1/\delta))(\log^7 d)(2 \log d + \log(1/\delta)) \log(nd)$$
$$= O(\log^9(nd/\delta)).$$

Thus, we complete the proof. $\square$

Here, we present a QUERY for outputting all the $n$ distances. In Section E, we provide a more general version, which can take any input set $\mathsf{S} \subseteq [n]$, and output distance for only them in a shorter time that proportional to $|\mathsf{S}|$.

**Lemma 4.3** (QUERY time, informal). *Given a query point $q \in \mathbb{R}^d$, the procedure* QUERY *(Algorithm 7) runs in time*

$$O((d + n \cdot \mathrm{mmc}(l)^2) \cdot \mathrm{poly}(1/\epsilon, \log(nd/\delta))).$$

*Proof.* The QUERY operation has the following two parts:

- **Part 1:** Line 5 takes $O(RLU \cdot \mathcal{T}_{\mathrm{encode}})$ time to call ENCODE to generate sketches for $q$.

- **Part 2:** For every $i \in [n]$:
    - Line 13 takes $O(RLU \cdot \mathcal{T}_{\mathrm{subtract}})$ time to compute sketch of the difference between $q$ and $x_i$, and store the sketch at index of $n + 1$.
    - Line 14 takes $O(RLU \cdot \mathcal{T}_{\mathrm{decode}})$ time to decode the BATCHHEAVYHITTER and get estimated heavy hitters of $q - x_i$.
    - Line 16 to Line 24 takes $O(RLU \cdot 2/\beta)$ time to analyze the BATCHHEAVYHITTER and get the set of indices, where $2/\beta$ is the size of the set.
    - Line 30 takes $O(LP \cdot 2/\beta)$ time to compute size of the layer sets cut by $\alpha$.
    - Line 32 to Line 36 takes $O(PL)$ time to compute the estimation of each layer.

  The total running time of this part is:

  $$n \cdot (O(RLU \cdot \mathcal{T}_{\mathrm{subtract}}) + O(RLU \cdot \mathcal{T}_{\mathrm{decode}}) + O(RLU \cdot 2/\beta) + O(LP \cdot 2/\beta) + O(LP))$$
  $$= O(nL(RU(\mathcal{T}_{\mathrm{subtract}} + \mathcal{T}_{\mathrm{decode}} + \beta^{-1}) + P\beta^{-1}))$$

  time in total.

Taking these two parts together we have the total running time of the QUERY procedure:

$$O(RLU \cdot \mathcal{T}_{\mathrm{encode}}) + O(nL(RU(\mathcal{T}_{\mathrm{subtract}} + \mathcal{T}_{\mathrm{decode}} + \beta^{-1}) + P\beta^{-1}))$$
$$= O(RLU(\mathcal{T}_{\mathrm{encode}} + n \cdot \mathcal{T}_{\mathrm{subtract}} + n \cdot \mathcal{T}_{\mathrm{decode}} + n\beta^{-1}) + nLP\beta^{-1})$$
$$= O(\epsilon^{-4} \log^6(d/\delta) \log(nd)(d \log^2(d) + n\beta^{-1} \log^2(d)) + n\epsilon^{-1} \log^2(d)\beta^{-1})$$
$$= O((d + n \cdot \mathrm{mmc}(l)^2) \cdot \mathrm{poly}(1/\epsilon, \log(nd/\delta)))$$

where the first step follows from the property of big $O$ notation, the second step follows from the definition of $R, L, U, \mathcal{T}_{\mathrm{encode}}, \mathcal{T}_{\mathrm{encode}}, \mathcal{T}_{\mathrm{subtract}}, \mathcal{T}_{\mathrm{decode}}$ (Theorem D.3), $P$, the third step follows from merging the terms.

Thus, we complete the proof.

$\square$

Next, we analyze the space usage in our algorithm. We sketch the proof in below and delay the full proof into Section G.

**Lemma 4.4** (Space complexity of our data structure, informal version of Lemma G.1). *Our data structure (Algorithm 1 and 2) uses space*

$$O(n(d + \mathrm{mmc}(l)^2) \cdot \mathrm{poly}(1/\epsilon, \log(d/\delta))).$$

*Proof Sketch.* First, we store the original data with space for $x = O(nd)$.

Second, we store the sub stream/sample of original data with

$$\text{space for } \overline{x} = \widetilde{O}(\epsilon^{-4} nd).$$

Third, our data structure holds a set $\{S_{r,l,u}\}_{r\in[R],l\in[L],u\in[U]}$, each $S_{r,l,u}$ of size of $\widetilde{O}(n \cdot \beta^{-1})$. We show that total space needed is

$$\text{space for } S = \widetilde{O}(\epsilon^{-9} n \cdot \mathrm{mmc}(l)^2).$$

Forth, we hold a bit-map $\mathrm{bmap}$ of size $O(RLUd)$, which uses the space of

$$\text{space for } \mathrm{bmap} = \widetilde{O}(\epsilon^{-4} d).$$

Fifth, in QUERY procedure, we generate a set of sets $\{H_{r,l,u}\}_{r\in[R],l\in[L],u\in[U]}$, each of the sets has size of $O(\beta^{-1})$, so the whole set uses space of

$$\text{space for } H = \widetilde{O}(\epsilon^{-9} \cdot \mathrm{mmc}(l)^2).$$

By putting them together, the total space complexity is

$$\text{space for } x + \text{space for } \overline{x} + \text{space for } S + \text{space for } \mathrm{bmap} + \text{space for } H$$
$$= O(nd) + \widetilde{O}(\epsilon^{-4} nd) + \widetilde{O}(\epsilon^{-9} n \cdot \mathrm{mmc}(l)^2) + \widetilde{O}(\epsilon^{-4} d) + \widetilde{O}(\epsilon^{-9} \cdot \mathrm{mmc}(l)^2)$$
$$= \widetilde{O}(\epsilon^{-9} n(d + \mathrm{mmc}(l)^2)).$$

$\square$

## 5 Correctness of Our Algorithm

The correctness of our distance oracle is proved in the following lemma:

**Lemma 5.1** (QUERY correctness). *Given a query point $q \in \mathbb{R}^d$, the procedure QUERY (Algorithm 7) takes $q$ as input and approximately estimates $\{\mathrm{dst}_i\}_{i\in[n]}$ the distance between $q$ and every $x_i$ with the norm $l$, such that for every $\mathrm{dst}_i$, with probability at least $1 - \delta$, we have*

$$(1 - \epsilon) \cdot \|q - x_i\|_{\mathrm{sym}} \leq \mathrm{dst}_i \leq (1 + \epsilon) \cdot \|q - x_i\|_{\mathrm{sym}}$$

*Proof.* Without loss of generality, we can consider a fixed $i \in [n]$. For simplicity, we denote $x_i$ by $x$.

Let $v := q - x$. By Lemma C.3, it is approximated by its layer vector $\mathcal{L}(v)$, namely,

$$\|v\|_{\mathrm{sym}} \leq \|\mathcal{L}(v)\|_{\mathrm{sym}} \leq (1 + O(\epsilon))\|v\|_{\mathrm{sym}}, \tag{1}$$

where $\| \cdot \|_{\mathrm{sym}}$ is a symmetric norm, denoted also by $l(\cdot)$.

We assume without loss of generality that $\epsilon \geq 1/\mathrm{poly}(d)$. Our algorithm maintains a data structure that eventually produces a vector $\mathcal{J}(v)$, which is created with the layer sizes $c_1, c_2, \ldots, c_P$, where $c$'s denotes the estimated layer sizes output by out data structure, and the $b$'s are the ground truth layer sizes. We will show that with high probability, $\|\mathcal{J}(v)\|_{\mathrm{sym}}$ approximates $\|\mathcal{L}(v)\|_{\mathrm{sym}}$. Specifically, to achieve $(1 \pm \epsilon)$-approximation to $\|v\|_{\mathrm{sym}}$, we set the approximation guarantee of the layer sets (Definition C.1) to be $\epsilon_1 := O(\frac{\epsilon^2}{\log(d)})$ and the importance guarantee to be $\beta_0 := O(\frac{\epsilon^5}{\mathrm{mmc}(l)^2 \log^5(d)})$, where $\mathrm{mmc}(l)$ is defined as Definition 1.3.

Observe that the number of non-empty layer sets $P = O(\log_\alpha(d)) = O(\log(d)/\epsilon)$. Let $E_{\text{succeed}}$ denote the event $(1 - \epsilon_1)b_k \leq c_k \leq b_k$. By Lemma F.1, it happens with high probability $1 - \delta$. Conditioned on this event.

Denote by $\mathcal{J}(v)$ the vector generated with the layer sizes given by Line 35, and by $\mathcal{L}^*(v)$ the vector $\mathcal{L}(v)$ after removing all buckets (Definition C.2) that are not $\beta$-contributing (Definition C.5), and define $\mathcal{J}^*(v)$ similar to $\mathcal{L}^*(v)$, where we set $\beta := \epsilon/P = O(\epsilon^2/\log(d))$. Every $\beta$-contributing layer is necessarily $\beta_0$-important (Definition C.1) by Lemma C.7 and Lemma C.8 and therefore satisfies $c_k \geq (1 - \epsilon_1)b_k$ (Lemma F.1). We bound the error of $\|\mathcal{L}^*(v)\|_{\text{sym}}$ by Lemma C.6, namely,

$$(1 - O(\epsilon))\|\mathcal{L}(v)\|_{\text{sym}} \leq (1 - O(\log_\alpha d \cdot \beta))\|\mathcal{L}(v)\|_{\text{sym}} \leq \|\mathcal{L}^*(v)\|_{\text{sym}} \leq \|\mathcal{L}(v)\|_{\text{sym}}.$$

where the first step follows from the definition of $\beta$, the second step and the third step follow from Lemma C.6.

Then, we have

$$
\begin{aligned}
\|\mathcal{J}(v)\|_{\text{sym}} &\geq \|\mathcal{J}^*(v)\|_{\text{sym}} \\
&= \|\mathcal{L}^*(v) \backslash \mathcal{L}_{k_1}(v) \cup \mathcal{J}_{k_1}(v) \ldots \backslash \mathcal{L}_{k_\kappa}(v) \cup \mathcal{J}_{k_\kappa}(v)\|_{\text{sym}} \\
&\geq (1 - \epsilon_1)^P \|\mathcal{L}^*(v)\|_{\text{sym}} \\
&\geq (1 - O(\epsilon))\|\mathcal{L}^*(v)\|_{\text{sym}}.
\end{aligned}
\tag{2}
$$

where the first step follows from monotonicity (Lemma B.1), the second step follows from the definition of $\mathcal{J}^*(v)$, the third step follows from Lemma C.4, and the fourth step follows from the definition of $\epsilon_1$ and $P$.

Combining Eq. (1) and (2), we have

$$(1 - O(\epsilon)) \cdot \|v\|_{\text{sym}} \leq \|\mathcal{J}^*(v)\|_{\text{sym}} \leq \|v\|_{\text{sym}}, \tag{3}$$

which bounds the error of $\|\mathcal{J}^*(v)\|_{\text{sym}}$ as required. Note that, with Lemma C.6 we have

$$(1 - O(\epsilon)) \cdot \|\mathcal{J}(v)\|_{\text{sym}} \leq \|\mathcal{J}^*(v)\|_{\text{sym}} \leq \|\mathcal{J}(v)\|_{\text{sym}} \tag{4}$$

Combining the Eq.(3) and (4) we have

$$(1 - O(\epsilon)) \cdot \|v\|_{\text{sym}} \leq \|\mathcal{J}(v)\|_{\text{sym}} \leq (1 + O(\epsilon)) \cdot \|v\|_{\text{sym}}$$

Note that $E_{\text{succeed}}$ has a failure probability of $\delta$. Thus, we complete the proof. $\square$

# 6 Conclusion

Similarity search is the backbone of many large-scale applications in machine-learning, optimization, databases and computational geometry. Our work strengthens and unifies a long line of work on metric embeddings and sketching, by presenting the first Distance Oracle *for any symmetric norm*, with nearly-optimal query and update times. The generality of our data structure allows to apply it as a *black-box* for data-driven *learned* symmetric distance metrics [DKJ+07] and in various optimization problems involving symmetric distances.

Our work raises several open questions for future study:

- The efficiency of our data structure depends on $\text{mmc}(l)$, the concentration property of the symmetric norm. Is this dependence necessary?
- Can we generalize our data structure to certain *asymmetric norms*?
- Can we extend our symmetric-norm distance oracle to any (non-linear) metric space (i.e., graphs)?

We believe our work is also likely to influence other fundamental problems in high-dimensional optimization and search, e.g, kernel linear regression, geometric sampling and near-neighbor search.

## Acknowledgments

RZ was supported by the University Graduate Continuing Fellowship from UT Austin.

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
