**Roadmap.** We divide the appendix into the following sections. Section A gives the preliminaries for our work. Section B introduces some useful properties of symmetric norms. Section C gives the proofs for the layer approximation technique. Section D states the formal version of our main theorem and algorithms. Section E gives more details about the time complexity proofs. Section F gives some details about the correctness proofs. Section G shows the space complexity of our algorithm. Section H states a streaming lower bound for the norm estimation problem and shows that our result is tight in this case. Section I gives an instantiation of the sparse recovery data structure (BATCHHEAVYHITTER) that simplifies the analysis in prior work.

## A Preliminaries

In Section A.1, we define the notations we use in this paper. In Section A.2, we introduce some probability tools. In Section A.3, we define $p$-stable distributions.

### A.1 Notations

For any positive integer $n$, we use $[n]$ to denote a set $\{1, 2, \cdots, n\}$. We use $\mathbb{E}[]$ to denote the expectation. We use $\Pr[]$ to denote the probability. We use $\mathrm{Var}[]$ to denote the variance. We define the unit vector $\xi^{(d_1)} := \frac{1}{\sqrt{d_1}}(1, 1, 1, \ldots, 1, 0, \ldots, 0) \in \mathbb{R}^d$, for any $d_1 \in [d]$, which has $d_1$ nonzero coordinates. We abuse the notation to write $\xi^{(d_1)} \in \mathbb{R}^{d_1}$ by removing zero coordinates, and vice-versa by appending zeros. We use $\|x\|_2$ to denote entry-wise $\ell_2$ norm of a vector. We use $\|x\|_{\mathrm{sym}}$ to denote the symmetric norm of a vector $x$.

We define tail as follows

**Definition A.1** (Tail of a vector). *For a given $x \in \mathbb{R}^d$ and an integer $k$, we use $x_{\overline{[k]}}$ or $x_{\mathrm{tail}(k)}$ to denote the vector that without largest top-k values (in absolute).*

### A.2 Probability Tools

We state some useful inequalities in probability theory in below.

**Lemma A.2** (Chernoff bound [Che52]). *Let $X = \sum_{i=1}^n X_i$, where $X_i = 1$ with probability $p_i$ and $X_i = 0$ with probability $1 - p_i$, and all $X_i$ are independent. Let $\mu = \mathbb{E}[X] = \sum_{i=1}^n p_i$. Then*

- $\Pr[X \geq (1 + \delta)\mu] \leq \exp(-\delta^2 \mu/3), \forall \delta > 0$;

- $\Pr[X \leq (1 - \delta)\mu] \leq \exp(-\delta^2 \mu/2), \forall 0 < \delta < 1$.

**Lemma A.3** (Chebyshev's inequality). *Let $X$ be a random variable with finite expected value $\mu$ and finite non-zero variance $\sigma^2$. Then for any real number $k > 0$,*

$$\Pr[|X - \mu| \geq k\sigma] \leq \frac{1}{k^2}.$$

**Theorem A.4** (Levy's isoperimetric inequality, [GMS86]). *For a continuous function $f : S^{d-1} \to \mathbb{R}$, Let $M_f$ be the median of $f$, i.e., $\mu(\{x : f(x) \leq M_f\}) \geq 1/2$ and $\mu(\{x : f(x) \geq M_f\}) \geq 1/2$, where $\mu(\cdot)$ is the Haar probability measure on the unit sphere $S^{d-1}$. Then*

$$\mu(\{x : f(x) = M_f\}_\epsilon) \geq 1 - \sqrt{\pi/2}e^{-\epsilon^2 d/2},$$

*where for a set $A \subset S^{d-1}$ we denote $A_\epsilon := \{x : l_2(x, A) \leq \epsilon\}$ and $l_2(x, A) := \inf_{y \in A} \|x - y\|_2$.*

### A.3 Stable Distributions

We define $p$-stable distributions.

**Definition A.5** ([Ind06]). *A distribution $\mathcal{D}_p$ is called p-stable, if there exists $p \geq 0$ such that for any $d$ real numbers $a_1, a_2, \ldots, a_d$ and i.i.d. variables $x_1, x_2, \ldots, x_d$ from distribution $\mathcal{D}_p$. the random variable $\sum_{i=1}^d a_i x_i$ has the same distribution as the variable $\|a\|_p y$, where y is a random variable from distribution $\mathcal{D}_p$.*

# B Symmetric Norms

In this section, we give several technical tools for the symmetric norms. In Section B.1, we show the monotonicity of symmetric norms. In Section B.2, we show the concentration properties of symmetric norms. In Section B.3, we show some properties of the median of symmetric norm.

## B.1 Monotonicity Property of Symmetric Norm

**Lemma B.1** (Monotonicity of Symmetric Norms, see e.g. Proposition IV.1.1 in [Bha97] ). *If $\| \cdot \|_{\mathrm{sym}}$ is a symmetric norm and $x, y \in \mathbb{R}^d$ satisfy that for all $i \in [d]$, $|x_i| \leq |y_i|$, then $\|x\|_{\mathrm{sym}} \leq \|y\|_{\mathrm{sym}}$.*

## B.2 Concentration Property of Symmetric Norms

Let us give the results of concentration of measure as followed. The following tools and proofs can be found in [BBC$^+$17]. However, for the completeness, we state them in below.

**Lemma B.2** (Concentration of $M_l$). *For every norm $l$ on $\mathbb{R}^d$, if $x \in S^{d-1}$ is drawn uniformly at random according to Haar measure on the sphere, then*

$$\Pr[|\|x\|_l - M_l| > \frac{2\mathfrak{b}_l}{\sqrt{d}}] < \frac{1}{3}$$

*Proof.* With Theorem A.4, we know that, for a random $x$ distributed according to the Haar measure on the $l_2$-sphere, with probability at least $1 - \sqrt{\pi/2}e^{-2} > \frac{2}{3}$, we can always find some $y \in S^{d-1}$, such that

$$\|x - y\|_2 \leq \frac{2}{\sqrt{d}}, \text{ and}$$
$$\|y\|_l = M_l.$$

If we view the norm $l$ as a function, then it is obvious that it is $\mathfrak{b}_l$-Lipschitz with respect to $\| \cdot \|_2$, so that we have

$$\begin{aligned}
|\|x\|_l - M_l| &= |\|x\|_l - \|y\|_l| \\
&\leq \|x - y\|_l \\
&\leq \mathfrak{b}_l \|x - y\| \\
&\leq \frac{2\mathfrak{b}_l}{\sqrt{d}}
\end{aligned}$$

where the first step follows from the definition of $y$, the second step follows from triangle inequality, the third step follows from that norm $l$ is $\mathfrak{b}_l$-Lipschitz with respect to $\| \cdot \|_2$, and the last step follows from the definition of $y$.

Thus, we complete the proof. $\square$

**Lemma B.3** (Concentration inequalities for norms ). *For every $d > 0$, and norm $l$ on $\mathbb{R}^n$, there is a vector $x \in S^{d-1}$ satisfying*

- $|\|x\|_\infty - M_{l_\infty^{(d)}}| \leq 2/\sqrt{d}$

- $|\|x\|_l - M_{l^{(d)}}| \leq 2\mathfrak{b}_{l^{(d)}}/\sqrt{d}$, *and*

- $|\{i : |x_i| > \frac{1}{K\sqrt{d}}\}| > \frac{d}{2}$ *for some universal constant $K$.*

*Proof.* Let $x$ be drawn uniformly randomly from a unit sphere. From Lemma B.2, we have

$$\Pr[|\|x\|_l - M_{l_\infty^{(d)}}| > \frac{2}{\sqrt{d}}] < \frac{1}{3}$$
$$and \quad \Pr[|\|x\|_l - M_{l^{(d)}}| > \frac{2\mathfrak{b}_l}{\sqrt{d}}] < \frac{1}{3}.$$

Define $\tau(x, t) := |\{i \in [d] : |x_i| < t\}|$. Now we are giving the proof to show that, for a constant $K$, over a choice of $x$, we have $\tau(x, \frac{1}{K\sqrt{d}}) < \frac{d}{2}$ with probability lager than $2/3$ .

Let's consider an random vector $z \in \mathbb{R}^d$, such that each entry $z_i$ is independent standard normal random variable. It is well known that $\frac{z}{\|z\|_2}$ is distributed uniformly over a sphere, so it has the same distribution as $x$. There is a universal constant $K_1$ such that

$$\Pr[\|z\|_2 > K_1\sqrt{d}] < \frac{1}{6},$$

and similarly, there is a constant $K_2$, such that

$$\Pr[|z_i| < \frac{1}{K_2}] < \frac{1}{12}.$$

Therefore, by Markov bound we have

$$\Pr[\tau(z, \frac{1}{K_2}) > \frac{d}{2}] < \frac{1}{6}.$$

By union bound, with probability larger than $2/3$, it holds simultaneously that

$$\|z\|_2 \leq K_1\sqrt{d} \quad \text{and} \quad \tau(z, \frac{1}{K_2}) < \frac{d}{2},$$

which imply:

$$\tau(\frac{z}{\|z\|_2}, \frac{1}{K_1 K_2 \sqrt{d}}) < \frac{d}{2}.$$

Now, by union bound, a random vector $x$ satisfies all of the conditions in the statement of the lemma with positive probability.

Thus, we complete the proof. $\qquad\square$

### B.3 Median of Symmetric Norm

We state a well-known fact before introducing Lemma B.5.

**Fact B.4** (Concentration for median on infinity norm, [GMS86]). *There are absolute constants $0 < \gamma_1 < \gamma_2$ such that for every integer $d \geq 1$,*

$$\gamma_1 \sqrt{\log(d)/d} \leq M_{l_\infty^{(d)}} \leq \gamma_2 \sqrt{\log(d)/d}$$

We now give the following Lemma, which says that the $l$-norm of the (normalized) unit vector $\xi^{(d)}$ is closely related to the median of the norm. We are considering this vector because that it is related to a single layer of $\mathcal{L}$.

**Lemma B.5** (Flat Median Lemma ). *Let $l : \mathbb{R}^d \to \mathbb{R}$ be a symmetric norm. Then*

$$\lambda_1 M_l / \sqrt{\log(d)} \leq l(\xi^{(d)}) \leq \lambda_2 M_l$$

*where $\lambda_1, \lambda_2 > 0$ are absolute constants.*

We notice that the first inequality is tight for $l_\infty$.

*Proof.* Using Lemma B.3, we can find a vector $x \in S^{d-1}$ and a constant $\lambda > 0$ satisfying

- $|\|x\|_\infty - M_{l_\infty}| \leq \lambda\sqrt{1/d}$

- $|\|x\|_l - M_l| \leq \lambda \mathfrak{b}_l / \sqrt{d}$, and

- $|\{i : |x_i| > \frac{1}{K\sqrt{d}}\}| > \frac{d}{2}$ for some universal constant $K$.

With Fact B.4, $M_{l_\infty} = \Theta(\sqrt{\log(d)/d})$. On the other hand, let $\mathrm{mmc}(l)$ be defined as Definition 1.3, we have

$$\mathrm{mmc}(l) \leq \gamma\sqrt{d}$$

for sufficiently small $\gamma$, thus

$$\frac{\lambda\mathfrak{b}_l}{\sqrt{d}} < M_l.$$

So with constants $\gamma_1, \gamma_2 > 0$, we have

$$\gamma_1 M_l \leq \|x\|_l \leq \gamma_2 M_l$$
$$and \quad \gamma_1\sqrt{\log(d)/d} \leq \|x\|_\infty \leq \gamma_2\sqrt{\log(d)/d}.$$

So that we have

$$|x| \leq \gamma_2\sqrt{\log(d)} \cdot \xi^{(d)},$$

where the inequality is entry-wise, and with monotonicity of symmetric norms (Lemma B.1), we have

$$\gamma_1 \leq \|x\|_l \leq \gamma_2\sqrt{\log(d)} \cdot \|\xi^{(d)}\|_l.$$

Now we move to the second condition of the lemma, we first let $M = \{i : |x_i| > \frac{1}{K\sqrt{d}}\}$. As $|M| > \frac{d}{2}$, we can find a permutation $\pi$ satisfying

$$[d] - M \in \pi(M).$$

We define a vector $\pi(x)$ to be the vector by applying the permutation $\pi$ to each entry of $x$. Denote by $|x|$ the vector of taking absolute value of $x$ entry-wise. Notice $|x| + \pi(|x|) > \frac{\xi^{(d)}}{K}$ entry-wise, so that we have

$$\begin{aligned}
\frac{1}{K} \cdot \|\xi^{(d)}\|_l &\leq \||x| + \pi(|x|)\|_l \\
&\leq \||x|\|_l + \|\pi(|x|)\|_l \\
&= 2\|x\|_l \\
&\leq 2\gamma_2 M_l,
\end{aligned}$$

where the first step follows from the monotonicity of symmetric norm (Lemma B.1), the second step follows from the triangle inequality, the third step follows from the definition of $\pi$, and the last step follows from the definition of $\gamma_2$.

Thus, we complete the proof. $\qquad\square$

The following lemma shows the monotonicity of the median (in $d$), a very useful property in the norm approximation.

**Lemma B.6** (Monotonicity of Median). *Let $l : \mathbb{R}^d \to \mathbb{R}$ be a symmetric norm. For all $d_1 \leq d_2$, where $d_1, d_2 \in [n]$, let $\mathrm{mmc}(l)$ be defined as Definition 1.3, we have*

$$M_{l^{(d_1)}} \leq \lambda \cdot \mathrm{mmc}(l) \cdot \sqrt{\log(d_1)} M_{l^{(d_2)}},$$

*where $\lambda > 0$ is an absolute constant.*

*Proof.* By Lemma B.5 and the fact that $\xi^{(d_1)}$ is also a vector in $S^{d_2-1}$,

$$\lambda M_{l^{(d_1)}}/\sqrt{\log(d_1)} \leq \|\xi^{(d_1)}\|_l \leq \mathfrak{b}_{l^{(d_2)}} \leq \mathrm{mmc}(l) M_{l^{(d_2)}}.$$

Thus, we complete the proof. $\qquad\square$

# C  Analysis of Layer Approximation

In this section, we show how to estimate the symmetric norm $l(\cdot)$ of a vector using the layer vectors. Section C.1 gives the definitions of layer vectors and important layers. Section C.2 shows that we can approximate the exact value of the norm and the layer vector. Section C.3 defines the contributing layer and shows its concentration property. Section C.4 proves that contributing layers are also important.

Throughout this section, let $\epsilon \in (0,1)$ be the precision, $\alpha > 0$ and $\beta \in (0,1]$ be some parameters depending on $d$, $\epsilon$ and $\mathrm{mmc}(l)$, where $\mathrm{mmc}(l)$ is defined as Definition 1.3. Furthermore, we assume $\mathrm{mmc}(l) \leq \gamma\sqrt{d}$, for constant parameter $0 \leq \gamma \ll 1/2$ small enough.[2]

## C.1  Layer Vectors and Important Layers

**Definition C.1** (Important Layers). *For $v \in \mathbb{R}^d$, define layer $i \in \mathbb{N}_+$ as*

$$B_i := \{j \in [d] : \alpha^{i-1} < |v_j| \leq \alpha^i\},$$

*and denote its size by $b_i := |B_i|$. We denote the number of non-zero $b_i$'s by $t$, the number of non-empty layers. And we say that layer-$i$ is $\beta$-important if*

- $b_i > \beta \cdot \sum_{j=i+1}^{t} b_j$

- $b_i \alpha^{2i} \geq \beta \cdot \sum_{j \in [i]} b_j \alpha^{2j}$

With out loss of generality, We restrict the entries of the vector $v$ to be in $[-m, m]$, and that $m = \mathrm{poly}(d)$. Then we know that the number of non-zero $b_i$'s is at most $P = O(\log_\alpha(d))$. In the view of $\ell_2$-norm, for an arbitrary vector $v \in \mathbb{R}^d$, if we normalize it to a unit vector, then the absolute value of each non-zero entry is at least $1/\mathrm{poly}(d)$. In order to simplify our analysis and algorithm for approximating $\|v\|_{\mathrm{sym}}$, we introduce the notations we use as follows.

**Definition C.2** (Layer Vectors and Buckets). *For each $i \in [P]$, let $\alpha^i \cdot \mathbf{1}_{b_i} \in \mathbb{R}^{b_i}$ denote a vector that has length $b_i$ and every entry is $\alpha^i$. Define the layer vector for $v \in \mathbb{R}^d$ with integer coordinates to be*

$$\mathcal{L}(v) := (\alpha^1 \cdot \mathbf{1}_{b_1}, \alpha^2 \cdot \mathbf{1}_{b_2}, \cdots, \alpha^P \cdot \mathbf{1}_{b_P}, 0 \cdot \mathbf{1}_{d - \sum_{j \in [P]} b_j}) \in \mathbb{R}^d;$$

*and define the $i$-th bucket of $\mathcal{L}(v)$ to be*

$$\mathcal{L}_i(v) := (0 \cdot \mathbf{1}_{b_1 + b_2 + \cdots + b_{i-1}}, \alpha^i \cdot \mathbf{1}_{b_i}, 0 \cdot \mathbf{1}_{d - \sum_{j \in [i]} b_j}) \in \mathbb{R}^d;$$

*We also define $\mathcal{J}(v)$ and $\mathcal{J}_i(v)$ as above by replacing $\{b_i\}$ with the approximated values $\{c_i\}$. Denote $\mathcal{L}(v) \backslash \mathcal{L}_i(v)$ as the vector with the $i$-th bucket of $\mathcal{L}(v)$ replaced by 0. We also denote $(\mathcal{L}(v) \backslash \mathcal{L}_i(v)) \cup \mathcal{J}_i(v)$ as the vector by replacing the $i$-th bucket of $\mathcal{L}(v)$ with $\mathcal{J}_i(v)$, i.e.,*

$$(\mathcal{L}(v) \backslash \mathcal{L}_i(v)) \cup \mathcal{J}_i(v) := (\alpha^1 \cdot \mathbf{1}_{b_1}, \alpha^2 \cdot \mathbf{1}_{b_2}, \cdots, \alpha^i \cdot \mathbf{1}_{c_i}, \cdots, \alpha^P \cdot \mathbf{1}_{b_P}, 0 \cdot \mathbf{1}_{d - \sum_{j \in [P]} b_j + b_i - c_i}) \in \mathbb{R}^d;$$

## C.2  Approximated Layers Provides a Good Norm Approximation

We now proof that, $\|v\|_{\mathrm{sym}}$ can be approximated by using layer vector $V$. We first choose a base to be $\alpha := 1 + O(\epsilon)$.

**Lemma C.3** (Approximattion with Layer Vector). *For all $v \in \mathbb{R}^d$, we have*

$$\|\mathcal{L}(v)\|_{\mathrm{sym}}/\alpha \leq \|v\|_{\mathrm{sym}} \leq \|\mathcal{L}(v)\|_{\mathrm{sym}}.$$

*Proof.* The lemma follows from the monotonicity of symmetric norms(Lemma B.1) directly.  □

The next key lemma shows that $\|\mathcal{J}(v)\|_{\mathrm{sym}}$ is a good approximation to $\|\mathcal{L}(v)\|_{\mathrm{sym}}$.

**Lemma C.4** (Bucket Approximation ). *For every layer $i \in [P]$,*

---

[2]We note that beyond this regime, the streaming lower bound in Theorem H.2 implies that a linear-sized memory (time) is required to approximate the norm.

- *if $c_i \le b_i$, then $\|(\mathcal{L}(v)\backslash\mathcal{L}_i(v)) \cup \mathcal{J}_i(v)\|_{\mathrm{sym}} \le \|\mathcal{L}(v)\|_{\mathrm{sym}}$;*
- *if $c_i \ge (1-\epsilon)b_i$, then $\|(\mathcal{L}(v)\backslash\mathcal{L}_i(v)) \cup \mathcal{J}_i(v)\|_{\mathrm{sym}} \ge (1-\epsilon)\|\mathcal{L}(v)\|_{\mathrm{sym}}$.*

*Proof.* With the monotonicity of norm (Lemma B.1), the upper bound is quite obvious. So we just focus on the lower bound. Let us take the vector

$$\mathcal{J}_i(v) := (0 \cdot \mathbf{1}_{c_1+c_2+\cdots+c_{i-1}}, \alpha^i \cdot \mathbf{1}_{c_i}, 0, \cdots, 0) \in \mathbb{R}^d;$$

Here we define $\mathcal{K}(v) := \mathcal{L}(v) - \mathcal{L}_i(v)$. Then notice that $\mathcal{K}(v) + \mathcal{J}_i(v)$ is a permutation of the vector $(\mathcal{L}(v)\backslash\mathcal{L}_i(v)) \cup \mathcal{J}_i(v)$. We will then show that, under assumptions of the lemma, we have

$$\|\mathcal{K}(v) + \mathcal{J}_i(v)\|_{\mathrm{sym}} \ge (c_i/b_i)\|\mathcal{L}(v)\|_{\mathrm{sym}}.$$

Assume a vector $v \in \mathbb{R}^d$ and a permutation $\pi \in \Sigma_d$, we define a vector $\pi(x)$ to be the vector by applying the permutation $\pi$ to each entry of $x$. Using the property of the symmetric norm, we have that $\|v\|_{\mathrm{sym}} = \|\pi(v)\|_{\mathrm{sym}}$. Consider a set of permutations that are cyclic shifts over the non-zero coordinates of $\mathcal{L}_i$, and do not move any other coordinates. That is, there is exactly $b_i$ permutations in $S$, and for every $\pi \in S$, we have $\pi(\mathcal{K}(v)) = \mathcal{K}(v)$. By the construction of $S$, we have,

$$\sum_{\pi \in S} \pi(\mathcal{J}_i(v)) = c_i\mathcal{L}_i(v)$$

and therefore $\sum_{\pi \in S} \pi(\mathcal{K}(v) + \mathcal{J}_i(v)) = c_i\mathcal{L}_i(v) + b_i\mathcal{K}(v)$. As the vectors $\mathcal{L}_i(v)$ and $\mathcal{K}(v)$ have disjoint support, by monotonicity of symmetric norm (Lemma B.1) with respect to each coordinates we can deduce $\|c_i\mathcal{L}_i(v) + b_i\mathcal{K}(v)\|_{\mathrm{sym}} \ge \|c_i(\mathcal{L}_i(v) + \mathcal{K}(v))\|_{\mathrm{sym}}$. By plugging those together,

$$
\begin{aligned}
c_i\|\mathcal{L}_i(v) + \mathcal{K}(v)\|_{\mathrm{sym}} &\le \|c_i\mathcal{L}_i(v) + b_i\mathcal{K}(v)\|_{\mathrm{sym}} \\
&= \|\sum_{\pi \in S} \pi(\mathcal{J}_i(v) + \mathcal{K}(v))\|_{\mathrm{sym}} \\
&\le \sum_{\pi \in S} \|\pi(\mathcal{J}_i(v) + \mathcal{K}(v))\|_{\mathrm{sym}} \\
&= b_i\|\pi(\mathcal{J}_i(v) + \mathcal{K}(v))\|_{\mathrm{sym}}
\end{aligned}
$$

where the first step follows from $c_i\|\mathcal{L}_i(v) + \mathcal{K}(v)\|_{\mathrm{sym}} = \|c_i(\mathcal{L}_i(v) + \mathcal{K}(v))\|_{\mathrm{sym}}$ and the monotonicity of the norm $l$, the second step follows from $\sum_{\pi \in S} \pi(\mathcal{J}_i(v) + \mathcal{K}(v)) = c_i\mathcal{L}_i(v) + b_i\mathcal{K}(v)$, the third step follows from triangle inequality, and the last step follows from the property of symmetric norm and $|S| = b_i$.

Hence,

$$\|\mathcal{J}_i(v) + \mathcal{K}(v)\|_{\mathrm{sym}} \ge \frac{c_i}{b_i}\|\mathcal{L}(v)\|_{\mathrm{sym}} \ge (1-\epsilon)\|\mathcal{L}(v)\|_{\mathrm{sym}},$$

Thus, we complete the proof. $\square$

### C.3 Contributing Layers

**Definition C.5** (Contributing Layers). *For $i \in [P]$, layer $i$ is called $\beta$-contributing if*

$$\|\mathcal{L}_i(v)\|_{\mathrm{sym}} \ge \beta\|\mathcal{L}(v)\|_{\mathrm{sym}}.$$

**Lemma C.6** (Concentration with contributing layers). *Let $\mathcal{L}^*(v)$ be the vector obtained from V by removing all layers that are not $\beta$-contributing. Then*

$$(1 - O(\log_\alpha(d)) \cdot \beta) \cdot \|\mathcal{L}(v)\|_{\mathrm{sym}} \le \|\mathcal{L}^*(v)\|_{\mathrm{sym}} \le \|\mathcal{L}(v)\|_{\mathrm{sym}}.$$

*Proof.* Let $i_1, \ldots, i_k \in [P]$ be the layers that are not $\beta$-contributing.

Then we apply the triangle inequality and have,

$$
\begin{aligned}
\|\mathcal{L}(v)\|_{\mathrm{sym}} &\ge \|\mathcal{L}(v)\|_{\mathrm{sym}} - \|\mathcal{L}_{i_1}(v)\|_{\mathrm{sym}} - \cdots - \|\mathcal{L}_{i_k}(v)\|_{\mathrm{sym}} \\
&\ge (1 - k_\beta)\|\mathcal{L}(v)\|_{\mathrm{sym}}
\end{aligned}
$$

The proof follows by bounding $k$ by $P = O(\log_\alpha(n))$, which is the total number of non-zero $b_i$'s. $\square$

## C.4 Contributing Layers Are Important

In this section, we give two lemmas to show that every $\beta$-contributing layer (Definition C.5) is $\beta'$-important (Definition C.1), where $\beta'$ is depending on $\mathrm{mmc}(l)$ (Definition 1.3). The first property of the important layer is proved in Lemma C.7, and the second property is proved in Lemma C.8.

**Lemma C.7** (Importance of contributing layers (Part 1)). *For $i \in [P]$, if layer $i$ is $\beta$-contributing, then for some absolute constant $\lambda > 0$, we have*

$$b_i \geq \frac{\lambda \beta^2}{\mathrm{mmc}(l)^2 \log^2(d)} \cdot \sum_{j=i+1}^{P} b_j,$$

*where $\mathrm{mmc}(l)$ is defined as Definition 1.3.*

*Proof.* We first fix a layer $i$ which is $\beta$-contributing. Let $\mathcal{U}(v)$ be the vector $\mathcal{L}(v)$ after removing buckets $j = 0, \ldots, i$. By Lemma B.5, there is an absolute constant $\lambda_1 > 0$ such that

$$\begin{aligned}
\|\mathcal{L}_i(v)\|_{\mathrm{sym}} &= \alpha^i \cdot \sqrt{b_i} \cdot l(\xi^{(b_i)}) \\
&\leq \lambda_1 \cdot \alpha^i \cdot \sqrt{b_i} \cdot M_{l^{(b_i)}},
\end{aligned}$$

and similarly

$$\|\mathcal{U}(v)\|_{\mathrm{sym}} \geq \frac{\lambda_2 \alpha^i}{\sqrt{\log(d)}} \cdot \left( \sum_{j=i+1}^{P} b_j \right)^{1/2} M_{l^{(\Sigma_{j=i+1}^{P} b_j)}}.$$

With these two inequalities, we can have the following deduction.

First, we have

$$\|\mathcal{L}_i(v)\|_{\mathrm{sym}} \geq \beta \cdot \|\mathcal{L}(v)\|_{\mathrm{sym}} \geq \beta \cdot \|\mathcal{U}(v)\|_{\mathrm{sym}}.$$

Second, we assume that $b_i < \sum_{j=i+1}^{P} b_j$, as otherwise we are done:

$$b_i \geq \sum_{j=i+1}^{P} b_j \geq \frac{\lambda \beta^2}{\mathrm{mmc}(l)^2 \log^2(d)} \cdot \sum_{j=i+1}^{P} b_j.$$

Then, by the monotonicity of the median (Lemma B.6), we have

$$M_{l^{(b_i)}} \leq \lambda_3 \cdot \mathrm{mmc}(l) \cdot \sqrt{\log(d)} \cdot M_{l^{(\Sigma_{j=i+1}^{P} b_j)}}$$

for some absolute constant $\lambda_3 > 0$.

Putting it all together, we get

$$\beta \cdot \frac{\lambda_2 \alpha^i}{\sqrt{\log(d)}} \cdot \left( \sum_{j=i+1}^{P} b_j \right)^{1/2} \leq \lambda_1 \cdot \alpha^i \sqrt{b_i} \cdot \lambda_3 \cdot \mathrm{mmc}(l) \cdot \sqrt{\log(d)}.$$

Therefore, we finish the proof. $\square$

**Lemma C.8** (Importance of contributing layers (Part 2)). *For a symmetric $l$, let $\mathrm{mmc}(l)$ be defined as Definition 1.3. If layer $i \in [P]$ is $\beta$-contributing, then there is an absolute constant $\lambda > 0$ such that*

$$b_i \alpha^{2i} \geq \frac{\lambda \beta^2}{\mathrm{mmc}(l)^2 \cdot \log_\alpha(n) \cdot \log^2(n)} \cdot \sum_{j \in [i]} b_j \alpha^{2j}.$$

*Proof.* We first fix a layer $i$ which is $\beta$-contributing, and let $h := \arg\max_{j \leq i} \sqrt{b_j} \alpha^j$. We consider the two different cases as follows.

First, if $b_i \geq b_h$ then the lemma follows obviously by

$$\sum_{j \in [i]} b_j \alpha^{2j}$$
$$\leq t \cdot b_h \cdot \alpha^{2h}$$
$$\leq O(\log_\alpha(d)) \cdot b_i \cdot \alpha^{2i}.$$

The second case is when $b_i < b_h$. With Definition C.5 and Lemma B.5, we have

$$\lambda_1 \cdot \alpha^i \cdot \sqrt{b_i} \cdot M_{l^{(b_i)}}$$
$$\geq \|\mathcal{L}_i\|_{\mathrm{sym}}$$
$$\geq \beta \cdot \|\mathcal{L}\|_{\mathrm{sym}}$$
$$\geq \lambda_2 \cdot \beta \cdot \alpha^h \cdot \sqrt{\frac{b_h}{\log(d)}} \cdot M_{l^{(b_h)}},$$

for some absolute constants $\lambda_1, \lambda_2 > 0$, where the first step follows from Lemma B.5, the second step follows from Definition C.5, and the last step follows from Lemma B.5.

following from monotonicity of the median (Lemma B.6), we can plugging in $M_{l^{(b_i)}} \leq \lambda_3 \cdot \mathrm{mmc}(l) \cdot \sqrt{\log(d)} \cdot M_{l^{(b_h)}}$, for some absolute constant $\lambda_3 > 0$, so that we have

$$\lambda_1 \cdot \alpha^i \cdot \sqrt{b_i} \cdot M_{l^{(b_i)}} \geq \frac{\lambda_2 \cdot \beta \cdot \sqrt{b_h} \cdot \alpha^h}{\sqrt{\log(d)}} \cdot \frac{M_{l^{(b_i)}}}{\lambda_3 \cdot \mathrm{mmc}(l) \cdot \sqrt{\log(d)}},$$
$$\sqrt{b_i} \cdot \alpha^i \geq \frac{\lambda_2 \cdot \beta \cdot \sqrt{b_h} \cdot \alpha^h}{\lambda_1 \lambda_3 \cdot \mathrm{mmc}(l) \cdot \log(d)}.$$

Square the above inequality and we can see that $b_h \cdot \alpha^{2h} \geq \frac{1}{O(\log_\alpha(d))} \cdot \sum_{j \in [i]} b_j \alpha^{2j}$.

Thus we complete the proof. $\qquad\square$

## D  Formal Main Result and Algorithms

In this section, we state the formal version of our main theorem and algorithms. Section D.1 presents our main result: a data structure for distance estimation with symmetric norm. Section D.2 introduces the sparse recovery tools for sketching.

### D.1  Formal Version of Our Main Result

---
**Algorithm 3** Data structure for symmetric norm estimation

---
1: **data structure** DISTANCEONSYMMETRICNORM                                          ▷ Theorem D.1
2:
3: **private:**
4: **procedure** LAYERVECTORAPPROX($\alpha, b_1, b_2, \ldots, b_P, d$)                   ▷ Lemma C.3
5:     For each $i \in [P]$, let $\alpha^i \cdot \mathbf{1}_{b_i} \in \mathbb{R}^{b_i}$ denote a vector that has length $b_i$ and every entry is $\alpha^i$
6:     $\mathcal{L} \leftarrow (\alpha^1 \cdot \mathbf{1}_{b_1}, \cdots, \alpha^P \cdot \mathbf{1}_{b_P}, 0, \ldots, 0) \in \mathbb{R}^d$       ▷ Generate the layer vector
7:     **return** $\|\mathcal{L}\|_{\mathrm{sym}}$               ▷ Return the norm of the estimated layer vector
8: **end procedure**
9: **end data structure**

---

Here we divide function QUERY presented in Theorem 1.4 into two versions. One takes only the query point $q \in \mathbb{R}^d$ as input to ask all the distances. Another takes a query point $q \in \mathbb{R}^d$ and a set $S \subset [n]$ to ask for the distance with the specific set of points. The latter can be viewed as a more general version of the former. In the former parts, we have proved the correctness of the query (Lemma 5.1), and the running time of version for all points (Lemma 4.3). Now we state the both in the following theorem, and the running time analysis for the latter will be stated in Section E.

---

**Algorithm 4** Data structure for symmetric norm estimation: members, formal version of Algorithm 1

---

1: **data structure** DISTANCEONSYMMETRICNORM        ▷ Theorem D.1
2: **members**
3:     $d, n \in \mathbb{N}_+$        ▷ $n$ is the number of points, $d$ is dimension
4:     $X = \{x_i \in \mathbb{R}^d\}_{i=1}^n$        ▷ Set of points being queried
5:     $L \in \mathbb{N}_+$        ▷ number of layers we subsample
6:     $R \in \mathbb{N}_+$        ▷ number of substreams in one layer
7:     $\epsilon, \delta$
8:     $\beta$        ▷ used to cut important layer
9:     $U \in \mathbb{N}_+$        ▷ number of parallel processing
10:     BATCHHEAVYHITTER $\{S_{r,l,u}\}_{r \in [R], l \in [L], u \in [U]}$
11:     $\{H_{r,l,u} \subset [d] \times \mathbb{R}\}_{r \in [R], l \in [L], u \in [U]}$    ▷ each set $H$ has a size of $2/\beta$, and is used to store the output of BATCHHEAVYHITTER
12:     $\gamma$        ▷ parameter used when cutting layer vector
13:     bmap $\in \{0,1\}^{R \times L \times U \times d}$
14:     $\overline{x}_{r,l,u} \in \mathbb{R}^{n \times d}$, for each $r \in [R], l \in [L], u \in [U]$        ▷ Substreams
15: **end members**
16: **end data structure**

---

**Algorithm 5** Data structure for symmetric norm estimation: init, formal version of Algorithm 1

---

1: **data structure** DISTANCEONSYMMETRICNORM        ▷ Theorem D.1
2: **public:**
3: **procedure** INIT($\{x_1, \cdots, x_i\} \subset \mathbb{R}^d, n \in \mathbb{N}_+, d \in \mathbb{N}_+, \delta \in (0, 0.1), \epsilon \in (0, 0.1)$)   ▷ Lemma 4.1
4:     $n \leftarrow n, d \leftarrow d, \delta \leftarrow \delta, \epsilon \leftarrow \epsilon$
5:     **for** $i = 1 \rightarrow n$ **do**
6:        $x_i \leftarrow x_i$
7:     **end for**
8:     $\epsilon_1 \leftarrow O(\frac{\epsilon^2}{\log d})$        ▷ We define this notation for purpose of analysis
9:     $L \leftarrow \log(d), R \leftarrow \Theta(\epsilon_1^{-2} \log(n/\delta) \log^2 d), U \leftarrow \lceil \log(nd^2/\delta) \rceil$
10:     $\beta \leftarrow O(\frac{\epsilon^5}{\mathrm{mmc}(l)^2 \log^5 d})$
11:     **for** $r \in [R], l \in [L], u \in [U]$ **do**
12:        $S_{r,l,u}$.INIT($\sqrt{\beta}, n+2, d$)        ▷ Theorem D.3
13:     **end for**
14:     **for** $r \in [R], u \in [U], j \in [d], l \in [L]$ **do**
15:        Draw $\xi \in [0, 1]$
16:        **if** $\xi \in [0, 2^{-l}]$ **then**
17:           bmap$[r, l, u, j] \leftarrow 1$
18:        **else**
19:           bmap$[r, l, u, j] \leftarrow 0$
20:        **end if**
21:     **end for**
22:     **for** $r \in [R], u \in [U], i \in [n], j \in [d], l \in [L]$ **do**
23:        **if** bmap$[r, l, u, j] = 1$ **then**
24:           $S_{r,l,u}$.ENCODESINGLE($i, j, x_{i,j}, d$)        ▷ Theorem D.3
25:           $[\overline{x}_{r,l,u}]_{i,j} \leftarrow x_{i,j}$        ▷ Create a copy of subvectors
26:        **else**
27:           $[\overline{x}_{r,l,u}]_{i,j} \leftarrow 0$
28:        **end if**
29:     **end for**
30: **end procedure**
31: **end data structure**

---

**Theorem D.1** (Main result, formal version of Theorem 1.4). *There is a data structure (Algorithm 4, 5, 7, 6) uses $O(\epsilon^{-9} n(d + \mathrm{mmc}(l)^2) \log^{14}(nd/\delta))$ spaces for the Online Approximate Adaptive Symmetric Norm Distance Estimation Problem (Definition 1.1) with the following procedures:*

---

**Algorithm 6** Data structure for symmetric norm estimation: update

---

1: **data structure** DISTANCEONSYMMETRICNORM          ▷ Theorem D.1
2: **public:**
3: **procedure** UPDATE($i \in [n], z \in \mathbb{R}^d$)          ▷ Lemma 4.2
4:                                  ▷ You want to replace $x_i$ by $z$
5:      **for** $r \in [R], u \in [U], j \in [d], l \in [L]$ **do**
6:          **if** $\mathrm{bmap}[r, l, u] = 1$ **then**
7:              $S_{r,l,u}$.ENCODESINGLE($i, j, z_j, d$)          ▷ Theorem D.3
8:              $[\overline{x}_{r,l,u}]_{i,j} \leftarrow z_j$          ▷ Create a copy of subvectors
9:          **else**
10:              $[\overline{x}_{r,l,u}]_{i,j} \leftarrow 0$
11:          **end if**
12:      **end for**
13: **end procedure**
14: **end data structure**

---

- INIT($\{x_1, x_2, \ldots, x_n\} \subset \mathbb{R}^d, \epsilon \in (0,1), \delta \in (0,1)$): *Given $n$ data points $\{x_1, x_2, \ldots, x_n\} \subset \mathbb{R}^d$, an accuracy parameter $\epsilon$ and a failure probability $\delta$ as input, the data structure preprocesses in time $O(\epsilon^{-9} n(d + \mathrm{mmc}(l)^2) \log^{14}(nd/\delta))$.*

- UPDATEX($z \in \mathbb{R}^d, i \in [n]$): *Given an update vector $z \in \mathbb{R}^d$ and index $i \in [n]$, the UPDATEX takes $z$ and $i$ as input and updates the data structure with the new $i$-th data point in $O(\epsilon^{-4} d \log^9(nd/\delta))$ time.*

- QUERY($q \in \mathbb{R}^d$) *(Querying all points): Given a query point $q \in \mathbb{R}^d$, the QUERY operation takes $q$ as input and approximately estimates the symmetric norm distances from $q$ to all the data points $\{x_1, x_2, \ldots, x_n\} \subset \mathbb{R}^d$ in time*

$$O(\epsilon^{-9}(d + n \cdot \mathrm{mmc}(l)^2) \log^{14}(nd/\delta))$$

  *i.e. it provides a set of estimates $\{\mathrm{dst}_i\}_{i=1}^n$ such that:*

$$\forall i \in [n], (1 - \epsilon)\|q - x_i\|_{\mathrm{sym}} \leq \mathrm{dst}_i \leq (1 + \epsilon)\|q - x_i\|_{\mathrm{sym}}$$

  *with probability at least $1 - \delta$.*

- QUERY($q \in \mathbb{R}^d, \mathsf{S} \subseteq [n]$) *(Querying a specific set $\mathsf{S}$ of points). Given a query point $q \in \mathbb{R}^d$ and an index $i \in [n]$, the QUERYONE operation takes $q$ and $i$ as input and approximately estimates the symmetric norm distances from $q$ to the $i$-th point $x_i \in \mathbb{R}^d$ in time*

$$O(\epsilon^{-9}(d + |\mathsf{S}| \cdot \mathrm{mmc}(l)^2) \log^{14}(nd/\delta))$$

  *i.e. it provides a estimated distance $\mathrm{dst} \in \mathbb{R}^{\mathsf{S}}$ such that:*

$$(1 - \epsilon)\|q - x_i\|_{\mathrm{sym}} \leq \mathrm{dst}_i \leq (1 + \epsilon)\|q - x_i\|_{\mathrm{sym}}, \forall i \in \mathsf{S}$$

  *with probability at least $1 - \delta$.*

- ESTPAIR($i, j \in [n]$) *Given indices $i, j \in [n]$, the ESTPAIR operation takes $i$ and $j$ as input and approximately estimates the symmetric norm distances from $i$-th to the $j$-th point $x_i, x_j \in \mathbb{R}^d$ in time*

$$O(\epsilon^{-9} \cdot \mathrm{mmc}(l)^2 \log^{14}(nd/\delta))$$

  *i.e. it provides a estimated distance* pair *such that:*

$$(1 - \epsilon)\|x_i - x_j\|_{\mathrm{sym}} \leq \mathrm{pair} \leq (1 + \epsilon)\|x_i - x_j\|_{\mathrm{sym}}$$

  *with probability at least $1 - \delta$.*

*Proof.* In Lemma E.3, Lemma E.4, Lemma E.1 and Lemma E.2 we analyze the running time for INIT, UPDATE and QUERY and ESTPAIR respectively.

**Algorithm 7** Data structure for symmetric norm estimation: query, formal version of Algorithm 2

```
 1: data structure DISTANCEONSYMMETRICNORM                                              ▷ Theorem D.1
 2: procedure QUERY(q ∈ ℝᵈ)                                                            ▷ Lemma 4.3, 5.1
 3:     P ← O(log_α(d))                                  ▷ P denotes the number of non-empty layer sets
 4:     for r ∈ [R], l ∈ [L], u ∈ [U] do
 5:         S_{r,l,u}.ENCODE(n + 1, q, d)                                          ▷ Generate Sketch for q
 6:     end for
 7:     ξ ← chosen uniformly at random from [1/2, 1]
 8:     γ ← Θ(ε)
 9:     α ← 1 + γ · ξ
10:     P ← O(log_α(d)) = O(log(d)/ε)                    ▷ P denotes the number of non-empty layer sets
11:     for i ∈ [n] do
12:         for r ∈ [R], l ∈ [L], u ∈ [U] do
13:             S_{r,l,u}.SUBTRACT(n + 2, n + 1, i)
14:             H_{r,l,u} ← S_{r,l,u}.DECODE(n + 2, √β, d)            ▷ This can be done in (2/β) poly(log d)
15:                                  ▷ At this point H_{r,l,u} is a list of index, the value of each index is reset to 0
16:             for k ∈ H_{r,l,u} do
17:                 value ← [x̄_{r,l,u}]_{i,k}
18:                 H_{r,l,u}[k] ← value
19:                 w ← ⌈log(value)/log(α)⌉
20:                 if α^{w-1} ≥ value/(1 + ε) then
21:                     H_{r,l,u} ← null
22:                     break
23:                 end if
24:             end for
25:             if H_{r,l,u} ≠ null then
26:                 H_{r,l} ← H_{r,l,u}
27:             end if
28:         end for
29:         for l ∈ [L], k ∈ [P] do
30:             A^i_{l,k} ← |{k | ∃k ∈ H_{r,l}, α^{k-1} < |H_{r,l}[k]| ≤ α^k}|
31:         end for
32:         for k ∈ [P] do
33:             q^i_k ← max_{l∈[L]} {l | A^i_{l,k} ≥ (R log(1/δ))/(100 log(d))}          ▷ Definition F.7
34:             If q^i_k does not exist, then η̂^i_k ← 0; Else η̂^i_k ← A_{q^i_k,k}/(R(1+ε₁))
35:             If η̂^i_k = 0 then c^i_k ← 0; Else c^i_k ← log(1-η̂^i_k)/(1 - w^{-q_k})
36:         end for
37:         dst_i ← LAYERVETCORAPPROX(α, c^i_1, c^i_2, ..., c^i_P, d)
38:         for r ∈ [R], l ∈ [L], u ∈ [U] do
39:             {H_{r,l,u}} ← {0}                                        ▷ Reset the sets to use for next point
40:         end for
41:     end for
42:     return {dst_i}_{i∈[n]}
43: end procedure
44: end data structure
```

Lemma G.1 shows the space complexity.

In Lemma 5.1 we give the correctness of QUERY, and correctness for ESTPAIR follows directly.

Thus, putting them all together, we prove the Theorem.

$\square$

## D.2 Sparse Recovery tools

We start with describing a data structure problem

**Definition D.2** (Batch Heavy Hitter). *Given an $n \times d$ matrix, the goal is to design a data structure that supports the following operations:*

- INIT($\epsilon \in (0, 0.1), n, d$). *Create a set of Random Hash functions and all the $n$ copies of sketches share the same hash functions.*

- ENCODE($i \in [n], z \in \mathbb{R}^d, d$). *This step encodes $z$ into $i$-th sketched location and store a size $\mathcal{S}_{\text{space}}$ linear sketch.*

- ENCODESINGLE($i \in [n], j \in [d], z \in \mathbb{R}, d$). *This step updates one sparse vector $e_j z \in \mathbb{R}^d$ into $i$-th sketched location.*

- SUBTRACT($i, j, l \in [n]$). *This function updates the sketch at $i$-th location by $j$-th sketch minus $l$-th sketch.*

- DECODE($i \in [n], \epsilon \in (0, 0.1), d$). *This function returns a set $L \subseteq [d]$ of size $|L| = O(\epsilon^{-2})$ containing all $\epsilon$-heavy hitters $i \in [n]$ under $\ell_p$. Here we say $i$ is an $\epsilon$-heavy hitter under $\ell_2$ if $|x_i| \geq \epsilon \cdot \|x_{\overline{[\epsilon^{-2}]}}\|_2$ where $x_{\overline{[k]}}$ denotes the vector $x$ with the largest $k$ entries (in absolute value) set to zero. Note that the number of heavy hitters never exceeds $2/\epsilon^2$.*

The existing work [KNPW11, Pag13] implies the following result. However their proofs are very decent and complicated. We provide another data structure in Section I that significantly simplifies the analysis (by only paying some extra log factors). We believe it is of independent interest.

**Theorem D.3.** *There is (linear sketch) data structure BATCHHEAVYHITTER($\epsilon, n, d$) that uses $\mathcal{S}_{\text{space}}$ space that support the following operations:*

- INIT($\epsilon \in (0, 0.1), n, d$). *This step takes $\mathcal{T}_{\text{init}}(\epsilon, n, d)$ time.*

- ENCODE($i \in [n], z \in \mathbb{R}^d, d$). *This step takes $\mathcal{T}_{\text{encode}}(d)$ time.*

- ENCODESINGLE($i \in [n], j \in [d], z \in \mathbb{R}, d$). *This step takes $\mathcal{T}_{\text{encodesingle}}(d)$ time.*

- SUBTRACT($i, j, l \in [n]$). *This step takes $\mathcal{T}_{\text{encodesingle}}(d)$ time.*

- DECODE($i \in [n], \epsilon \in (0, 0.1), d$). *This step takes $\mathcal{T}_{\text{decode}}(\epsilon, d)$ time.*

*The running time of function can be summarize as*

- $\mathcal{S}_{\text{space}}(\epsilon, d) = n \cdot O(\epsilon^{-2} \log^2 d)$

- $\mathcal{T}_{\text{init}}(\epsilon, n, d) = n \cdot O(\epsilon^{-2} \log^2 d)$

- $\mathcal{T}_{\text{encode}}(d) = O(d \log^2(d))$

- $\mathcal{T}_{\text{encodesingle}}(d) = O(\log^2(d))$

- $\mathcal{T}_{\text{subtract}}(\epsilon, d) = O(\epsilon^{-2} \log^2 d)$

- $\mathcal{T}_{\text{decode}}(\epsilon, d) = O(\epsilon^{-2} \log^2 d)$

*Note that the succeed probability is at least $0.99$.*

We remark that, to boost the probability from constant to $1 - 1/\operatorname{poly}(nd)$ we just need to pay an extra $\log(nd)$ factor.

## E   More Details of the Time Complexity

In this section, we analyze the running time of the general version of QUERY. Lemma 4.3 is a special case of the following Lemma when $\mathsf{S} = [n]$.

**Lemma E.1** (QUERY time for general version). *Given a query point $q \in \mathbb{R}^d$ and a set $\mathsf{S} \subseteq [n]$, the procedure QUERY (Algorithm 8) runs in time*

$$O(\epsilon^{-9}(d + |\mathsf{S}| \cdot \operatorname{mmc}(l)^2) \log^{14}(nd/\delta)).$$

*Proof.* The QUERY operation for a stored vector (Algorithm 8) has the following two parts:

---

**Algorithm 8** Data structure for symmetric norm estimation: query set

---

1: **data structure** DISTANCEONSYMMETRICNORM           ▷ Theorem D.1
2: **procedure** QUERY($q \in \mathbb{R}^d$, $\mathsf{S} \subseteq [n]$)           ▷ Lemma 4.3, 5.1
3:      $P \leftarrow O(\log_\alpha(d))$           ▷ $P$ denotes the number of non-empty layer sets
4:      **for** $r \in [R], l \in [L], u \in [U]$ **do**
5:          $S_{r,l,u}$.ENCODE($n+1, q, d$)           ▷ Generate Sketch for $q$
6:      **end for**
7:      $\xi \leftarrow$ chosen uniformly at random from $[1/2, 1]$
8:      $\gamma \leftarrow \Theta(\epsilon)$
9:      $\alpha \leftarrow 1 + \gamma \cdot \xi$
10:      $P \leftarrow O(\log_\alpha(d)) = O(\log(d)/\epsilon)$           ▷ $P$ denotes the number of non-empty layer sets
11:      **for** $i \in \mathsf{S}$ **do**
12:          **for** $r \in [R], l \in [L], u \in [U]$ **do**
13:              $S_{r,l,u}$.SUBTRACT($n+2, n+1, i$)
14:              $H_{r,l,u} \leftarrow S_{r,l,u}$.DECODE($n+2, \sqrt{\beta}, d$)           ▷ This can be done in $\frac{2}{\beta}$ poly($\log d$)
15:                        ▷ At this point $H_{r,l,u}$ is a list of index, the value of each index is reset to 0
16:              **for** $k \in H_{r,l,u}$ **do**
17:                  value $\leftarrow [\overline{x}_{r,l,u}]_{i,k}$
18:                  $H_{r,l,u}[k] \leftarrow$ value
19:                  $w \leftarrow \lceil \log(\text{value})/\log(\alpha) \rceil$
20:                  **if** $\alpha^{w-1} \geq$ value$/(1+\epsilon)$ **then**
21:                      $H_{r,l,u} \leftarrow$ null
22:                      **break**
23:                  **end if**
24:              **end for**
25:              **if** $H_{r,l,u} \neq$ null **then**
26:                  $H_{r,l} \leftarrow H_{r,l,u}$
27:              **end if**
28:          **end for**
29:          **for** $l \in [L], k \in [P]$ **do**
30:              $A_{l,k}^i \leftarrow |\{k \mid \exists k \in H_{r,l}, \alpha^{k-1} < |H_{r,l}[k]| \leq \alpha^k\}|$
31:          **end for**
32:          **for** $k \in [P]$ **do**
33:              $q_k^i \leftarrow \max_{l \in [L]} \{l \mid A_{l,k}^i \geq \frac{R \log(1/\delta)}{100 \log(d)}\}$
34:              If $q_k^i$ does not exist, then $\widehat{\eta}_k^i \leftarrow 0$; Else $\widehat{\eta}_k^i \leftarrow \frac{A_{q_k^i,k}}{R(1+\epsilon_2)}$
35:              If $\widehat{\eta}_k^i = 0$ then $c_k^i \leftarrow 0$; Else $c_k^i \leftarrow \frac{\log(1-\widehat{\eta}_k^i)}{1-w^{-q_k}}$
36:          **end for**
37:          dst$_i \leftarrow$ LAYERVETCORAPPROX($\alpha, c_1^i, c_2^i, \ldots, c_P^i, d$)
38:          **for** $r \in [R], l \in [L], u \in [U]$ **do**
39:              $\{H_{r,l,u}\} \leftarrow \{0\}$           ▷ Reset the sets to use for next point
40:          **end for**
41:      **end for**
42:      **return** $\{\text{dst}_i\}_{i \in \mathsf{S}}$
43: **end procedure**
44: **end data structure**

---

- **Part 1:** Line 5 takes $O(RLU \cdot \mathcal{T}_{\text{encode}})$ time to call ENCODE to generate sketches for $q$.

- **Part 2:** For each $i \in \mathsf{S}$

  - Line 13 takes $O(RLU \cdot \mathcal{T}_{\text{subtract}})$ time to compute sketch of the difference between $q$ and $x_i$, and store the sketch at index of $n+2$.

  - Line 14 takes $O(RLU \cdot \mathcal{T}_{\text{decode}})$ time to decode the BATCHHEAVYHITTER and get estimated heavy hitters of $q - x_i$.

  - Line 16 to Line 24 takes $O(RLU \cdot 2/\beta)$ time to analyze the BATCHHEAVYHITTER and get the set of indices, where $2/\beta$ is the size of the set.

  - Line 30 takes $O(LP \cdot 2/\beta)$ time to compute size of the layer sets cut by $\alpha$.

  - Line 32 to Line 36 takes $O(PL)$ time to compute the estimation of each layer.

---

**Algorithm 9** Data structure for symmetric norm estimation: query pair

---

1: **data structure** DISTANCEONSYMMETRICNORM        ▷ Theorem D.1
2: **procedure** ESTPAIR($i \in [n], j \in [n]$)        ▷ Lemma 4.3, 5.1
3:      $P \leftarrow O(\log_\alpha(d))$        ▷ $P$ denotes the number of non-empty layer sets
4:      $\xi \leftarrow$ chosen uniformly at random from $[1/2, 1]$
5:      $\gamma \leftarrow \Theta(\epsilon)$
6:      $\alpha \leftarrow 1 + \gamma \cdot \xi$
7:      $P \leftarrow O(\log_\alpha(d)) = O(\log(d)/\epsilon)$        ▷ $P$ denotes the number of non-empty layer sets
8:      **for** $r \in [R], l \in [L], u \in [U]$ **do**
9:          $S_{r,l,u}.\text{SUBTRACT}(n+2, j, i)$
10:          $H_{r,l,u} \leftarrow S_{r,l,u}.\text{DECODE}(n+2, \sqrt{\beta}, d)$        ▷ This can be done in $\frac{2}{\beta} \text{poly}(\log d)$
11:                  ▷ At this point $H_{r,l,u}$ is a list of index, the value of each index is reset to 0
12:          **for** $k \in H_{r,l,u}$ **do**
13:              value $\leftarrow [\overline{x}_{r,l,u}]_{i,k}$
14:              $H_{r,l,u}[k] \leftarrow$ value
15:              $w \leftarrow \lceil \log(\text{value}) / \log(\alpha) \rceil$
16:              **if** $\alpha^{w-1} \geq$ value$/(1+\epsilon)$ **then**
17:                  $H_{r,l,u} \leftarrow$ null
18:                  **break**
19:              **end if**
20:          **end for**
21:          **if** $H_{r,l,u} \neq$ null **then**
22:              $H_{r,l} \leftarrow H_{r,l,u}$
23:          **end if**
24:      **end for**
25:      **for** $l \in [L], k \in [P]$ **do**
26:          $A_{l,k} \leftarrow |\{k \mid \exists k \in H_{r,l}, \alpha^{k-1} < |H_{r,l}[k]| \leq \alpha^k\}|$
27:      **end for**
28:      **for** $k \in [P]$ **do**
29:          $q_k \leftarrow \max_{l \in [L]} \{l \mid A_{l,k} \geq \frac{R \log(1/\delta)}{100 \log(d)}\}$
30:          If $q_k$ does not exist, then $\widehat{\eta}_k \leftarrow 0$; Else $\widehat{\eta}_k \leftarrow \frac{A_{q_k,k}}{R(1+\epsilon_2)}$
31:          If $\widehat{\eta}_k = 0$ then $c_k \leftarrow 0$; Else $c_k \leftarrow \frac{\log(1-\widehat{\eta}_k)}{1-w^{-q_k}}$
32:      **end for**
33:      dst $\leftarrow$ LAYERVETCORAPPROX($\alpha, c_1, c_2, \ldots, c_P, d$)
34:      **return** dst
35: **end procedure**
36: **end data structure**

---

The total running time of this part is:

$$|\mathsf{S}| \cdot (O(RLU \cdot \mathcal{T}_{\text{subtract}}) + O(RLU \cdot \mathcal{T}_{\text{decode}}) + O(RLU \cdot 2/\beta) + O(LP \cdot 2/\beta) + O(LP)))$$
$$= O(|\mathsf{S}| \cdot L(RU(\mathcal{T}_{\text{subtract}} + \mathcal{T}_{\text{decode}} + \beta^{-1}) + P\beta^{-1}))$$

time in total.

Taking these two parts together we have the total running time of the QUERY procedure:

$$O(RLU \cdot \mathcal{T}_{\text{encode}}) + O(|\mathsf{S}| \cdot L(RU(\mathcal{T}_{\text{subtract}} + \mathcal{T}_{\text{decode}} + \beta^{-1}) + P\beta^{-1}))$$
$$= O(RLU(\mathcal{T}_{\text{encode}} + |\mathsf{S}| \cdot (\mathcal{T}_{\text{subtract}} + \mathcal{T}_{\text{decode}} + \beta^{-1})) + |\mathsf{S}| \cdot LP\beta^{-1})$$
$$= O(\epsilon^{-4} \log^6(nd/\delta)(d \log^2(d) \log(nd) + |\mathsf{S}| \cdot \beta^{-1} \log^2(d) \log(nd)) + |\mathsf{S}| \cdot \epsilon^{-1} \log^2(d)\beta^{-1})$$
$$= O(\epsilon^{-9}(d + |\mathsf{S}| \cdot \text{mmc}(l)^2) \log^{14}(nd/\delta))$$

where the first step follows from the property of big $O$ notation, the second step follows from the definition of $R, L, U, \mathcal{T}_{\text{encode}}, \mathcal{T}_{\text{encode}}, \mathcal{T}_{\text{subtract}}, \mathcal{T}_{\text{decode}}$ (Theorem D.3) , $P$, the third step follows from merging the terms.

Thus, we complete the proof.

$\square$

**Lemma E.2** (ESTPAIR time). *Given a query point $q \in \mathbb{R}^d$, the procedure* ESTPAIR *(Algorithm 9) runs in time*

$$O(\epsilon^{-9} \cdot \mathrm{mmc}(l)^2 \log^{14}(nd/\delta)).$$

*Proof.* The ESTPAIR operation (Algorithm 9) has the following two parts:

- Line 9 takes $O(RLU \cdot \mathcal{T}_{\mathrm{subtract}})$ time to compute sketch of the difference between $x_i$ and $x_j$, and store the sketch at index of $n + 2$.

- Line 10 takes $O(RLU \cdot \mathcal{T}_{\mathrm{decode}})$ time to decode the BATCHHEAVYHITTER and get estimated heavy hitters of $x_i - x_j$.

- Line 12 to Line 20 takes $O(RLU \cdot 2/\beta)$ time to analyze the BATCHHEAVYHITTER and get the set of indices, where $2/\beta$ is the size of the set.

- Line 26 takes $O(LP \cdot 2/\beta)$ time to compute size of the layer sets cut by $\alpha$.

- Line 28 to Line 32 takes $O(PL)$ time to compute the estimation of each layer.

The total running time is:

$$O(RLU \cdot \mathcal{T}_{\mathrm{subtract}}) + O(RLU \cdot \mathcal{T}_{\mathrm{decode}}) + O(RLU \cdot 2/\beta) + O(LP \cdot 2/\beta) + O(LP)$$
$$= O(RLU(\mathcal{T}_{\mathrm{subtract}} + \mathcal{T}_{\mathrm{decode}} + \beta^{-1}) + LP\beta^{-1})$$
$$= O(\epsilon^{-4} \log^6(nd/\delta)(\beta^{-1} \log^2(d) \log(nd) + \epsilon^{-1} \log^2(d)\beta^{-1} \log(nd))$$
$$= O(\epsilon^{-9} \cdot \mathrm{mmc}(l)^2 \log^{14}(nd/\delta))$$

time in total, where the first step follows from the property of big $O$ notation, the second step follows from the definition of $R, L, U, \mathcal{T}_{\mathrm{encode}}, \mathcal{T}_{\mathrm{encode}}, \mathcal{T}_{\mathrm{subtract}}, \mathcal{T}_{\mathrm{decode}}$ (Theorem D.3), $P$, the third step follows from merging the terms.

Thus, we complete the proof.

$\square$

**Lemma E.3** (INIT time, formal version of Lemma 4.1). *Given data points $\{x_1, x_2, \ldots, x_n\} \subset \mathbb{R}^d$, an accuracy parameter $\epsilon > 0$, and a failure probability $\delta > 0$ as input, the procedure* INIT *(Algorithm 5) runs in time*

$$O(\epsilon^{-9}n(d + \mathrm{mmc}(l)^2) \log^{14}(nd/\delta)).$$

*Proof.* The INIT time includes these parts:

- Line 12 takes $O(RLU \cdot \mathcal{T}_{\mathrm{init}}(\sqrt{\beta}, n, d))$ to initialize sketches

- Line 17 to Line 19 takes $O(RUdL)$ to generate the bmap;

- Line 24 takes $O(ndRUL \cdot \mathcal{T}_{\mathrm{encodesingle}}(d))$ to generate sketches

By Theorems D.3, we have

- $\mathcal{T}_{\mathrm{init}}(\sqrt{\beta}, n, d)) = n \cdot O(\beta^{-1} \log^2(d) \log(nd)) = O(n \cdot \mathrm{mmc}(l)^2 \log^7(d) \log(nd)\epsilon^{-5})$,

- $\mathcal{T}_{\mathrm{encodesingle}}(d) = O(\log^2(d) \log(nd))$.

Adding them together we got the time of

$$O(RLU\mathcal{T}_{\mathrm{init}}(\sqrt{\beta}, n, d)) + O(RUdL) + O(ndRUL \cdot \mathcal{T}_{\mathrm{encodesingle}}(d))$$
$$= O(RLU(\mathcal{T}_{\mathrm{init}}(\sqrt{\beta}, n, d) + nd \cdot \mathcal{T}_{\mathrm{encodesingle}}(d)))$$
$$= O(\epsilon^{-4} \log(d/\delta) \log^4(d) \cdot \log(d) \cdot \log(nd^2/\delta) \log(nd)(n \cdot \mathrm{mmc}(l)^2 \log^7(d)\epsilon^{-5} + nd \log^2(d)))$$

$$= O(\epsilon^{-9} n (d + \mathrm{mmc}(l)^2) \log^{14}(nd/\delta)),$$

where the first step follows from merging the terms, the second step follows from the definition of $R, L, U, \mathcal{T}_{\mathrm{encodesingle}}(d), \mathcal{T}_{\mathrm{init}}$, the third step follows from merging the terms.

Thus, we complete the proof. □

**Lemma E.4** (UPDATE time, formal version of Lemma 4.2). *Given a new data point $z \in \mathbb{R}^d$, and an index $i$ where it should replace the original data point $x_i \in \mathbb{R}^d$. The procedure* UPDATE *(Algorithm 6) runs in time*

$$O(\epsilon^{-4} d \log^9(nd/\delta)).$$

*Proof.* The UPDATE operation calls BATCHHEAVYHITTER.ENCODE for $RLU$ times, so it has the time of

$$O(RLU \cdot \mathcal{T}_{\mathrm{encode}}(d)) = O(\epsilon^{-4} \log(n/\delta) \log^4(d) \cdot \log(d) \cdot \log(nd^2/\delta) \cdot d \log^2(d) \log(nd))$$
$$= O(\epsilon^{-4} d \log^9(nd/\delta))$$

where the first step follows from the definition of $R, L, U, \mathcal{T}_{\mathrm{encode}}(d)$, the second step follows from

$$\log(n/\delta) \log^4(d) \log(d) \log(nd^2/\delta) \log^2(d) \log(nd)$$
$$= (\log(n/\delta))(\log^7 d)(2 \log d + \log(n/\delta)) \log(nd)$$
$$= O(\log^9(nd/\delta))$$

Thus, we complete the proof. □

# F   More Details of the Correctness Proofs

In this section, we give the complete proofs of the correctness of our algorithms. In Section F.1 we state and prove the main result of this section, using the technical lemmas in the following subsections. In Section F.2, we define the trackable layers and show the connection with important layers (Definition C.1). In Section F.3, we analyze the sample probability and track probability of a layer. In Section F.4, we show that a good estimation of track probability implies a good approximation of the layer size, which completes the proof of the correctness.

## F.1   Correctness of Layer Size Estimation

We first show that, the estimation of layer sizes output by our data structure are good to approximate the exact values.

**Lemma F.1** (Correctness of layer size approximation). *We first show that, the layer sizes $c_1^i, c_2^i, \dots, c_P^i$ our data structure return satisfy*

- *for all $k \in P$, $c_k^i \leq b_k^i$;*
- *if $k$ is a $\beta$-important layer (Definition C.1) of $q - x_i$, then $c_k^i \geq (1 - \epsilon_1) b_k^i$,*

*with probability at least $1 - \delta$, where the $b_k^i$ is the ground truth $k$-th layer size of $q - x_i$.*

*Proof.* We first define two events as

- $E_1$: for all important layers $k \in [P]$, $q_k$ is well defined;
- $E_2$: for all $k \in [P]$, if $\widehat{\eta}_k > 0$ then $(1 - O(\epsilon)) \eta'_{k,q_k} \leq \widehat{\eta}_k \leq \eta'_{k,q_k}$.

With Lemma F.6 and Lemma F.8, we have that

$$\Pr[E_1 \cap E_2] \geq 1 - \delta^{O(\log(d))}.$$

When the output of every BATCHHEAVYHITTER is correct, if follows from Lemma F.5, Lemma F.6 and Lemma F.9 the algorithm outputs an approximation to the layer vector meeting the two criteria.

The BATCHHEAVYHITTER is used a total of $LR$ times, each with error probability at most $\delta/\mathrm{d}$. By a union bound over the layers, the failure probability is at most $(\mathrm{poly}(\log d)) \cdot \delta/d = o(\delta)$. Therefore, the total failure probability of the algorithm is at most $1 - o(\delta)$.

Thus, we complete the proof. $\qquad\square$

### F.2 Trackability of Layers

**Definition F.2** (Trackability of layers). *A layer $k \in [P]$ of a vector $x$ is $\beta$-trackable, if*

$$\alpha^{2k} \geq \beta \cdot \|x_{\overline{[\beta^{-1}]}}\|_2^2$$

*where $x_{\overline{[\kappa]}}$ is defined as Definition A.1.*

**Lemma F.3** (Importance and Trackability). *Let $\alpha$ be some parameter such that $\alpha \in [0, 2]$. Suppose subvector $\widetilde{x}$ is obtained by subsampling the original vector with probability $p$.*

*If $k \in [P]$ is a $\beta$-important layer, then for any $\lambda > P$, with probability at least $1 - P\exp(-\frac{\lambda pb_k}{P\beta})$, layer $k$ is $\frac{\beta}{\lambda pb_k}$-trackable.*

*In particular, if $pb_k = O(1)$, then with probability at least $1 - P\exp(-\Omega(\frac{\lambda}{P\beta}))$, layer $k$ is $\frac{\beta}{\lambda}$-trackable.*

*Proof.* Let $(\zeta_0, \zeta_1, \dots)$ be the new level sizes of the subvector $\widetilde{x}$ we sampled. Thus, for $k \in [P]$, we have

$$\mathbb{E}[\zeta_k] = p \cdot b_k.$$

By the definition of important layer (Definition C.1) we have

$$\mathbb{E}[\zeta_k] \geq \beta \cdot \mathbb{E}\Big[\sum_{j=k+1}^{P} \zeta_j\Big]$$

and

$$\mathbb{E}[\zeta_k \cdot \alpha^{2k}] \geq \beta \cdot \mathbb{E}\Big[\sum_{j \in [k]} \zeta_j \cdot \alpha^{2j}\Big].$$

To have layer $k$ trackable, it has to be in the top $\lambda pb_k/\beta$ elements, so that we have $\sum_{j=k+1}^{P} \zeta_j \leq \lambda pb_k/\beta$. By Chernoff bound (Lemma A.2), we can know the probability that this event does not happen is

$$\Pr\Big[\sum_{j=k+1}^{P} b_j > \frac{\lambda pb_k}{\beta}\Big] \leq \exp(-\Omega(\lambda pb_k/\beta)).$$

On the other hand, for level $k$ to be trackable $\alpha^{2k} \geq \frac{\beta}{\lambda pb_k}\sum_{j \in [k]} \zeta_j \alpha^{2j}$.

Thus, the complement occurs with probability

$$\Pr\Big[\frac{\beta}{\lambda pb_k}\sum_{j \in [k]} \zeta_j \alpha^{2j} > \alpha^{2k}\Big] \leq \Pr\Big[\exists j, \zeta_j \alpha^{2j} \geq \frac{\lambda pb_k \alpha^{2k}}{P\beta}\Big]$$

$$\leq \sum_{j \in [P]} \Pr\Big[\zeta_j \alpha^{2j} \geq \frac{\lambda pb_k \alpha^{2k}}{P\beta}\Big].$$

By Chernoff bound (Lemma A.2) and the fact that $\mathbb{E}[\zeta_j \alpha^{2j}] \leq pb_k \alpha^{2k}$, we have

$$\Pr\Big[\frac{\beta}{\lambda pb_k}\sum_{j \in [k]} \zeta_j \alpha^{2j} > \alpha^{2k}\Big] \leq P \cdot \exp\Big(-\Omega(\frac{\lambda pb_k \alpha^{2k}}{\alpha^{2j} P\beta})\Big)$$

$$\leq P \cdot \exp\Big(-\Omega(\frac{\lambda pb_k}{\beta})\Big).$$

Thus we complete the proof. $\qquad\square$

## F.3 Probability Analysis

We first define

**Definition F.4.** *For each $k \in [p]$, $l \in \mathbb{R}^+$, we define $\eta_{k,l} := 1 - (1 - p_l)^{b_k}$ to be the probability that at least one element from $B_k$ is sampled with the sampling probability of $p_l = 2^{-l}$.*

Set $\lambda = \Theta(P \log(1/\delta))$. Let $\eta_{k,l}^*$ be the probability that an element from $B_k$ is contained in $H_{1,l}$, such that, for $H_{r,l}$ with any other $r$, the probability is the same as $\eta_{k,l}^*$.

**Lemma F.5** (Sample Probability and Track Probability). *For any layer $k \in [P]$ we have $\eta_{k,l}^* < \eta_{k,l}$. Let $\delta$ and $\epsilon$ denote the two parameters such that $0 < \delta < \epsilon < 1$. If layer $k$ is a $\beta$-important layer and $p_l b_k = O(1)$, then*

$$\eta_{k,l}^* \geq (1 - \Theta(\epsilon)) \cdot \eta_{k,l}.$$

*Proof.* If one is in $H_{r,l}$, it has to be sampled, so we have $\eta_{k,l}^* \leq \eta_{k,l}$. On the other hand, using Lemma F.3, with probability at least $1 - t \exp(-\Omega(\frac{\lambda p b_k}{t\beta}))$, layer $k$ is $\frac{\beta}{\lambda p b_k}$-trackable.

We have

$$\frac{\beta}{\lambda p b_k} = \Theta(\frac{\beta}{P \log(1/\delta)})$$

where the last step follows from definition of $\lambda$.

Thus with probability at least

$$\eta_{k,l}(1 - \Theta(\delta)) \geq \eta_{k,l}(1 - O(\epsilon))$$

an element from $B_k$ is sampled and the element is reported by BATCHHEAVYHITTER.

Thus we complete the proof. $\qquad\square$

**Lemma F.6** (Probability Approximation). *For $k \in [P]$, let $\widehat{\eta}_k$ be defined as in Line 34 in Algorithm 7. With probability at least $1 - \delta^{\Omega(\log d)}$, for all $k \in [P]$, if $\widehat{\eta}_k \neq 0$ then*

$$(1 - O(\epsilon_1))\eta_{k,q_k}^* \leq \widehat{\eta}_k \leq \eta_{k,q_k}^*.$$

*Proof.* We define

$$\gamma := \frac{R \log(1/\delta)}{100 \log d}.$$

Recall the condition of Line 34, if $\widehat{\eta}_k \neq 0$, then we have

$$A_{q_k,k} \geq \gamma.$$

For a fixed $k \in [P]$, since the sampling process is independent for each $r \in [R]$, we can assume that

$$\mathbb{E}[A_{q_k,k}] = R\eta_{k,q_k} \geq \gamma.$$

Otherwise, by Chernoff bound (Lemma A.2), we get that

$$\Pr[A_{q_k,k} \geq \gamma] = o(\delta^{\Omega(\log d)}),$$

which implies that with probability at least $1 - \delta^{\Omega(\log d)}$, $\widehat{\eta}_k = 0$ for all $k \in [P]$, and we are done.

Thus, under this assumption, by Chernoff bound (Lemma A.2), we have

$$\Pr[|A_{q_k,k} - R \cdot \eta_{k,q_k}^*| \geq \epsilon R\eta_{k,q_k}] \leq \exp(-\Omega(\epsilon^2 \gamma)) = \delta^{\Omega(\log d)}.$$

Since $P = \text{poly} \log(d)$, by union bound over the $B_k$ , the event

$$(1 - \epsilon_1)\eta_{k,q_k} \leq \frac{A_{q_k,k}}{R_k} \leq (1 + \epsilon_1)\eta_{k,q_k}$$

holds for all $k \in [P]$ with probability at least $1 - \delta^{\Omega(\log(d))}$. Since $\widehat{\eta}_k = \frac{A_{k,q_k}}{R(1+\epsilon_1)}$ by definition, we get the desired bounds.

Thus we complete the proof. $\qquad\square$

**Definition F.7.** *We define $q_k$ as*

$$q_k := \max_{l \in [L]} \left\{ l \middle| A_{l,k} \geq \frac{R \log(1/\delta)}{100 \log(d)} \right\}.$$

*We say that $q_k$ is well-defined if the set in the RHS of the above definition is non-empty.*

At Line 33 in Algorithm 7, we define the $q_k^i$ for $i$-th vector as the definition above.

**Lemma F.8** (Maximizer Probability). *If layer $k \in [P]$ is important (Line 33 in Algorithm 7), then with probability at least $1 - \delta^{\Omega(\log d)}$, the maximizer $q_k$ is well defined.*

*Proof.* Lemma F.5 tells us that, when $p_k b_k = O(1)$, layer $k$ is at least $\Omega(\frac{\beta}{P \log(1/\delta)})$-trackable. On the other hand, if $P_k = 2^{-l_0} = 1/b_k$, we have $\eta_{k,l_0} = 1 - (1 - p_k)^{b_k} \geq 1/e$. Thus we have

$$\mathbb{E}[A_{l_0,k}] \geq R/e,$$

so by Chernoff bound (Lemma A.2),

$$\Pr[A_{l_0,k} \leq \frac{R \log(1/\delta)}{\log d}] \leq \exp(-\Omega(\frac{R \log(1/\delta)}{\log d}))$$
$$\leq \delta^{\Omega(\log d)}.$$

Thus we show that, there exists one $q_k \geq l_0$ with probability at least $1 - \delta^{\Omega(\log n)}$.

Since there are at most $P = \text{poly} \log(d)$ important layers, with probability at least $1 - \delta^{\Omega(\log n)}$, the corresponding value of $q_k$ is well defined for all important layers.

Thus we complete the proof. $\qquad\qquad\square$

### F.4 From Probability Estimation to Layer Size Approximation

The following lemma shows that, if we have a sharp estimate for the track probability of a layer, then we can obtain a good approximation for its size.

**Lemma F.9** (Track probability implies layer size approximation). *Suppose $q_k \geq 1$, $\epsilon \in (0, 1/2)$, and $d$ is sufficiently large. Define $\epsilon_1 := O(\epsilon^2/\log(d))$. We have for $k \in [P]$,*

- *Part 1. If $\widehat{\eta}_k \leq \eta_{k,q_k}$ then $c_k \leq b_k$.*

- *Part 2. If $\widehat{\eta}_i \geq (1 - \epsilon_1)\eta_{k,q_k}$ then $c_k \geq (1 - O(\epsilon_1))b_k$.*

*Proof.* We know that

$$b_k = \frac{\log(1 - \eta_{k,q_k})}{\log(1 - 2^{-q_k})},$$

which is a increasing function of $\eta_{k,q_k}$.

**Part 1.** If $\widehat{\eta}_k \leq \eta_{k,q_k}$, then $c_k \leq b_k$.

**Part 2.** If $\widehat{\eta}_k \geq (1 - O(\epsilon))\eta_{k,q_k}$ we have

$$c_k \geq b_k + \frac{\epsilon_1 \eta_{k,q_k}}{(1 - \eta_{k,q_k}) \log(1 - 2^{-q_k})} \geq b_k - O(\epsilon)b_k.$$

Thus we complete the proof. $\qquad\qquad\square$

## G Space Complexity

In this section, we prove the space complexity of our data structure.

**Lemma G.1** (Space complexity of our data structure, formal version of Lemma 4.4). *Our data structure (Algorithm 4 and 5) uses $O(\epsilon^{-9}n(d + \text{mmc}(l)^2) \log^{14}(nd/\delta))$ space.*

*Proof.* First, we store the original data,

$$\text{space for } x = O(nd).$$

Second, we store the sub stream/sample of original data

$$\text{space for } \overline{x} = O(RLUnd)$$
$$= O(\epsilon^{-4} \log(n/\delta) \log^4(d) \cdot \log(d) \cdot \log(nd^2/\delta) \cdot nd)$$
$$= O(\epsilon^{-4} nd \log^6(nd/\delta)).$$

Our data structure holds a set $\{S_{r,l,u}\}_{r \in [R], l \in [L], u \in [U]}$ (Line 10). Each $S_{r,l,u}$ has a size of $\mathcal{S}_{\text{space}}(\sqrt{\beta}, d) = O(n \cdot \beta^{-1} \log^2(d) \log(nd))$, which uses the space of

$$\text{space for } S = O(RLUn \cdot \beta^{-1} \log^2(d) \log(nd))$$
$$= O(\epsilon^{-4} \log(n/\delta) \log^4(d) \cdot \log(d) \cdot \log(nd) \cdot \log(nd^2/\delta) \cdot n \cdot (\epsilon^{-5} \cdot \text{mmc}(l)^2 \log^5(d)) \log^2(d))$$
$$= O(\epsilon^{-9} n \cdot \text{mmc}(l)^2 \log^{14}(nd/\delta))$$

where the first step follows from the definition of $R, L, U$, and the second step follows just simplifying the last step.

We hold a bmap (Line 17 and Line 19) of size $O(RLUd)$, which uses the space of

$$\text{space for bmap} = O(RLUd)$$
$$= O(\epsilon^{-4} \log(n/\delta) \log^4(d) \cdot \log(d) \cdot \log(nd^2/\delta) \cdot d)$$
$$= O(\epsilon^{-4} d \log^6(nd/\delta)).$$

In QUERY, we generate a set of sets $\{H_{r,l,u}\}_{r \in [R], l \in [L], u \in [U]}$, each of the sets has size of $O(\beta^{-1})$, so the whole set uses space of

$$\text{space for } H = O(RLU \cdot \beta^{-1})$$
$$= O(\epsilon^{-4} \log(n/\delta) \log^4(d) \cdot \log(d) \cdot \log(nd^2/\delta) \cdot \epsilon^{-5} \cdot \text{mmc}(l)^2 \log^5(d))$$
$$= O(\epsilon^{-9} \cdot \text{mmc}(l)^2 \log^{11}(nd/\delta)).$$

By putting them together, we have the total space is

$$\text{total space}$$
$$= \text{space for } x + \text{space for } \overline{x} + \text{space for } S + \text{space for bmap} + \text{space for } H$$
$$= O(nd) + O(\epsilon^{-4} nd \log^6(nd/\delta)) + O(\epsilon^{-9} n \cdot \text{mmc}(l)^2 \log^{13}(nd/\delta))$$
$$\quad + O(\epsilon^{-4} d \log^6(nd/\delta)) + O(\epsilon^{-9} \cdot \text{mmc}(l)^2 \log^{11}(nd/\delta))$$
$$= O(\epsilon^{-9} n(d + \text{mmc}(l)^2) \log^{14}(nd/\delta)).$$

Thus, we complete the proof.

$\square$

# H  Lower Bound From Previous Work

We first define turnstile streaming model (see page 2 of [LNNT16] as an example) as follows

**Definition H.1** (Turnstile streaming model)**.** *We define two different turnstile streaming models here:*

- *Strict turnstile: Each update $\Delta$ may be an arbitrary positive or negative number, but we are promised that $x_i \geq 0$ for all $i \in [n]$ at all points in the stream.*

- *General turnstile: Each update $\Delta$ may be an arbitrary positive or negative number, and there is no promise that $x_i \geq 0$ always. Entries in $x$ may be negative.*

Under the turnstile model, the *norm estimation problem* has the following streaming lower bound:

**Theorem H.2** (Theorem 1.2 in [BBC+17]). *Let $l$ be a symmetric norm on $\mathbb{R}^n$. Any turnstile streaming algorithm (Definition H.1) that outputs, with probability at least $0.99$, a $(1 \pm 1/6)$-approximation for $l(\cdot)$ must use $\Omega(\mathrm{mmc}(l)^2)$ bits of space in the worst case.*

We note that this problem is a special case of our symmetry norm distance oracle problem (i.e., with $n = 1$ and query vector $q = \mathbf{0}_d$ where $\mathbf{0}_d$ is a all zeros length-$d$ vector). And in this case, the query time of our data structure becomes $\widetilde{O}(\mathrm{mmc}(l)^2)$ for a constant-approximation, matching the streaming lower bound in Theorem H.2.

# I   Details About Sparse Recovery Tools

In this section we give an instantiation of the sparse recovery tool we use (Definition D.2) in Algorithms 10 - 12. Although the running times of our data structure are slightly worse (by some log factors) than the result of [KNPW11], it's enough for our symmetric norm estimation task. In terms of space requirement, the classical sparse recovery/compressed sensing only sublinear space is allowed. In our application, we're allowed to use linear space (e.g. $d$ per point, $nd$ in total). And more importantly, our instantiation has much simpler algorithm and analysis than the prior result.

---

**Algorithm 10** Our CountSketch for Batch heavy hitter

1: **data structure** BASICBATCHHEAVYHITTER                          $\triangleright$ Definition D.2
2: **members**
3:      $d, n \in \mathbb{N}^+$                        $\triangleright$ $n$ is the number of vectors, $d$ is the dimension.
4:      $\epsilon$                             $\triangleright$ We are asking for $\epsilon$-heavy hitters
5:      $\delta$                                $\triangleright$ $\delta$ is the failure probability
6:      $B$                  $\triangleright$ $B$ is the multiple number of each counter to take mean
7:      $L$               $\triangleright$ $L$ is the number of number of the level of the binary tree.
8:      $\eta$                     $\triangleright$ $\eta$ is the precision for $l_2$ norm estimation.
9:      $K$                     $\triangleright$ $K$ is the number of hash functions.
10:      $\{h_{l,k}\}_{l \in \{0,\ldots,L\}, k \in [K]} \subseteq [2^l] \times [B]$          $\triangleright$ hash functions.
11:      $\{C_{l,b}^i\}_{i \in [n], l \in \{0,\ldots,L\}, b \in [B]}$       $\triangleright$ The counters we maintain in CountSketch.
12:      $\{\sigma_{l,k}\}_{l \in \{0,\ldots,L\}, k \in [K]} \subseteq [d] \times \{+1, -1\}$    $\triangleright$ The hash function we use for norm estimation
13:      $\mathcal{Q}$             $\triangleright$ A instantiation of LPLPTAILESTIMATION (Algorithm 13)
14:      $\{\mathcal{D}_l\}_{l \in [L]}$             $\triangleright$ Instantiations of FPEST (Algorithm 14)
15: **end members**
16:
17: **public:**
18: **procedure** INIT($\epsilon, n \in \mathbb{N}^+, d \in \mathbb{N}^+, \delta$)
19:      $L \leftarrow \log_2 d$
20:      $\delta' \leftarrow \epsilon \delta / (12(\log d) + 1)$
21:      **for** $l \in \{0, \ldots, L\}$ **do**
22:          $\mathcal{D}_l$.INIT($n, d, l, 1/7, \epsilon, \delta'$)
23:      **end for**
24:      $C_0 \leftarrow$ greater than $1000$
25:      $\mathcal{Q}$.INIT($n, \epsilon^{-2}, 2, C_0, \delta$)
26: **end procedure**

---

The following lemma shows that the sparse recovery data structure satisfies our requirements in Theorem D.3.

## I.1   Our Sparse Recovery Tool

We first state the correctness, the proof follows from framework of [KNPW11].

**Lemma I.1.** *The function* DECODE($i, \epsilon, d, \delta$) *(Algorithm 12) returns a set $S \subseteq d$ of size $|S| = O(\epsilon^{-2})$ containing all $\epsilon$-heavy hitters of the $i$-column of the matrix under $l_2$ with probability of $1 - \delta$. Here we say $j$ is an $\epsilon$-heavy hitter under $l_2$ if $|x_j| \geq \epsilon \cdot \|x_{\overline{[\epsilon^{-2}]}}\|_2$ where $x_{\overline{[k]}}$ denotes the vector $x$ with the largest $k$ entries (in absolute value) set to zero. Note that the number of heavy hitters never exceeds $2/\epsilon^2$.*

---

**Algorithm 11** Our CountSketch for Batch heavy hitter

---

1: **data structure** BASICBATCHHEAVYHITTER                                    ▷ Definition D.2
2: **procedure** ENCODESINGLE($i \in [n], j \in [d], z \in \mathbb{R}, d$)
3:     **for** $l \in \{0, \dots, L\}$ **do**
4:         $\mathcal{D}_l$.UPDATE($i, j, z$)
5:     **end for**
6:     $\mathcal{Q}$.UPDATE($i, j, z$)
7: **end procedure**
8:
9: **procedure** ENCODE($i \in [n], z \in \mathbb{R}^d, d$)
10:     **for** $j \in [d]$ **do**
11:         ENCODESINGLE($i, j, z_j, d$)
12:     **end for**
13: **end procedure**
14:
15: **end data structure**

---

---

**Algorithm 12** Our CountSketch for Batch heavy hitter

---

1: **data structure** BASICBATCHHEAVYHITTER                                    ▷ Definition D.2
2: **procedure** SUBTRACT($i, j, k \in [n]$)
3:     **for** $l \in \{0, \dots, L\}$ **do**
4:         $\mathcal{D}_l$.SUBTRACT($i, j, k$)
5:     **end for**
6:     $\mathcal{Q}$.SUBTRACT($i, j, k$)
7: **end procedure**
8:
9: **procedure** DECODE($i \in [n], \epsilon, d$)
10:     EstNorm $\leftarrow \mathcal{Q}$.QUERY($i$)     ▷ Here the EstNorm is the estimated tail $l_2$-norm of $i$-vector.
11:     $S \leftarrow \{0\}, S' \leftarrow \emptyset$
12:     **for** $l \in \{0, \dots, L\}$ **do**                                ▷ The dyadic trick.
13:         **for** $\xi \in S$ **do**
14:             Est $\leftarrow \mathcal{D}$.QUERY($i, \xi$)
15:             **if** Est $\geq (3/4)\epsilon^2 \cdot$ EstNorm **then**
16:                 $S' \leftarrow S' \cup \{2\xi, 2\xi + 1\}$
17:             **end if**
18:         **end for**
19:         $S \leftarrow S', S' \leftarrow \emptyset$
20:     **end for**
21:     **return** $S$
22: **end procedure**
23: **end data structure**

---

*Proof.* The correctness follows from the framework of [KNPW11], and combining tail estimation (Lemma I.4) and norm estimation.

□

We next analyze the running time of our data structure in the following lemma:

**Lemma I.2.** *The time complexity of our data structure (Algorithm 10, Algorithm 11 and Algorithm 12) is as follows:*

- INIT *takes time of* $O(\epsilon^{-1}(n + d)\log^2(nd/\delta))$.

- ENCODESINGLE *takes time of* $O(\log^2(nd/\delta))$.

- ENCODE *takes time of* $O(d\log^2(nd/\delta))$.

- SUBTRACT *takes time of* $O(\epsilon^{-1}\log^2(nd/\delta))$.

---

**Algorithm 13** Batched $\ell_p$ tail estimation algorithm, based on [NS19]

---
1: **data structure** LpLpTailEstimation
2: **members**
3:   $m$                                                                           ▷ $m$ is the sketch size
4:   $\{g_{j,t}\}_{j\in[d],t\in[m]}$                            ▷ random variable that sampled i.i.d. from distribution $\mathcal{D}_p$
5:   $\{\delta_{j,t}\}_{j\in[d],t\in[m]}$                         ▷ Bernoulli random variable with $\mathbb{E}[\delta_{j,t}] = 1/(100k)$
6:   $\{y_{i,t}\}_{i\in[n],t\in[m]}$                                                              ▷ Counters
7: **end members**
8: **public:**
9: **procedure** Init($n, k, p, C_0, \delta$)
10:   $m \leftarrow O(\log(n/\delta))$
11:   Choose $\{g_{j,t}\}_{j\in[d],t\in[m]}$ to be random variable that sampled i.i.d. from distribution $\mathcal{D}_p$
12:   Choose $\{\delta_{j,t}\}_{j\in[d],t\in[m]}$ to be Bernoulli random variable with $\mathbb{E}[\delta_{j,t}] = 1/(100k)$  ▷ Matrix
    $A$ in Lemma I.4 is implicitly constructed based on $g_{j,t}$ and $\delta_{j,t}$
13:   initialize $\{y_{i,t}\}_{i\in[n],t\in[m]} = \{0\}$
14: **end procedure**
15:
16: **procedure** Update($i \in [n], j \in [d], z \in \mathbb{R}$)
17:   **for** $t \in [m]$ **do**
18:     $y_{i,t} \leftarrow y_{i,t} + \delta_{j,t} \cdot g_{jt} \cdot z$
19:   **end for**
20: **end procedure**
21:
22: **procedure** Subtract($i, j, k \in [n]$)
23:   **for** $t \in [m]$ **do**
24:     $y_{i,t} \leftarrow y_{j,t} - y_{k,t}$
25:   **end for**
26: **end procedure**
27:
28: **procedure** Query($i \in [n]$)
29:   $V \leftarrow \text{median}_{t\in[m]} |y_{i,t}|^2$
30:   **return** $V$
31: **end procedure**

---

- Decode *takes time of* $O(\epsilon^{-2} \log^2(nd/\delta))$.

*Proof.* We first notice that $L = \log_2 d$ and $\delta' = \epsilon\delta/(12(\log(d) + 1))$.

For the procedure Init (Algorithm 10), Line 22 takes time

$$O(L\epsilon^{-1} n \log(n/\delta')) = O(\epsilon^{-1} n \log^2(\delta^{-1}\epsilon^{-1} nd \log(d))).$$

Line 25 takes time $O((n + d) \log(n/\delta))$. Taking together, we have the total running time of Init is $O(\epsilon^{-1}(n + d) \log^2(\delta^{-1}\epsilon^{-1} nd \log(d))$

For the procedure EncodeSingle (Algorithm 11), Line 4 takes time of

$$O(L \log(n/\delta')) = O(\log^2(\epsilon^{-1}\delta^{-1} nd \log(d))).$$

Line 6 takes time of $O(\log(n/\delta))$. Taking together we have the total running time of EncodeSingle to be $O(\log^2(\epsilon^{-1}\delta^{-1} nd \log(d)))$.

For the procedure Encode (Algorithm 11), it runs EncodeSingle for $d$ times, so its running time is $O(d \log^2(\epsilon^{-1}\delta^{-1} nd \log(d)))$.

For the procedure Subtract (Algorithm 12), Line 4 runs in time

$$O(L\epsilon^{-1} \log(n/\delta')) = O(\epsilon^{-1} \log^2(\epsilon^{-1}\delta^{-1} nd \log(d))).$$

Line 6 runs in time $O(\log(n/\delta))$. So the total running time is $O(\epsilon^{-1} \log^2(\epsilon^{-1}\delta^{-1} nd \log(d)))$.

For the procedure DECODE (Algorithm 12), Line 10 runs in time $O(\log(n/\delta))$. Line 14 runs in time

$$O(L\epsilon^{-2}\log(n/\delta')) = O(\epsilon^{-2}\log^2(\epsilon^{-1}\delta^{-1}nd\log(d))).$$

So the total running time is $O(\epsilon^{-2}\log^2(\epsilon^{-1}\delta^{-1}nd\log(d)))$

Thus we complete the proof. □

The space complexity of our data structure is stated in below.

**Lemma I.3.** *Our batch heavy hitter data structure (Algorithm 10, Algorithm 11 and Algorithm 12) takes the space of*

$$O(\epsilon^{-1}(n+d)\log^2(\epsilon^{-1}\delta^{-1}nd\log(d))).$$

*Proof.* Our data structure has these two parts to be considered:

- The instantiations of FPEST we maintain.

- The instantiation of LPLPTAILESTIMATION we maintain .

The first part takes the space of

$$O(L\epsilon^{-1}n\log(n/\delta')) = O(\epsilon^{-1}n\log^2(\epsilon^{-1}\delta^{-1}nd\log(d))).$$

And the second part takes the space of $O(d\log(n/\delta))$. Adding them together we complete the proof. □

## I.2   Lp Tail Estimation

[NS19] provide a linear data structure LPLPTAILESTIMATION$(x, k, p, C_0, \delta)$ that can output the estimation of the contribution of non-heavy-hitter entries. We restate their Lemma as followed.

**Lemma I.4** (Lemma C.4 of [NS19])**.** *Let $C_0 \geq 1000$ denote some fixed constant. There is an oblivious construction of matrix $A \in \mathbb{R}^{m \times n}$ with $m = O(\log(1/\delta))$ along with a decoding procedure LPLPTAILESTIMATION$(x, k, p, C_0, \delta)$ such that, given $Ax$, it is possible to output a value $V$ in time O(m) such that*

$$\frac{1}{10k}\|x_{\overline{C_0 k}}\|_p^p \leq V \leq \frac{1}{k}\|x_{\overline{k}}\|_p^p,$$

*holds with probability $1 - \delta$.*

**Lemma I.5.** *The running time of the above data structure is*

- INIT *runs in time of $O((n+d)\log(n/\delta))$*

- UPDATE *runs in time of $O(\log(n/\delta))$*

- SUBTRACT *runs in time of $O(\log(n/\delta))$*

- QUERY *runs in time of $O(\log(n/\delta))$*

*And its space is $O(d\log(n/\delta))$.*

## I.3   Lp Norm Estimation

Following the work of [KNW10], we provide a linear sketch satisfying the following requirements.

**Lemma I.6** ([KNW10])**.** *There is an linear sketch data structure FPEST using space of*

$$O(\epsilon^{-1}\phi^{-2}n\log(n/\delta))$$

*and it provide these functions:*

- INIT$(n \in \mathbb{Z}^+, d \in \mathbb{Z}^+, l \in \mathbb{Z}^+, \phi, \epsilon, \delta)$: *Initialize the sketches, running in time $O(\epsilon^{-1}n\log(n/\delta))$*

---

**Algorithm 14** Batched $\ell_p$ norm estimation algorithm, based on [KNW10]

---

1: **data structure** LPNORMEST
2: **members**
3:     $R, T$                                                                  ▷ parallel parameters
4:     $m$                                                                          ▷ Sketch size
5:     $\{y_{r,t}^i\}_{i\in[n],r\in[R],t\in[T]} \subset \mathbb{R}^m$                            ▷ Sketch vectors
6:     $\{A\}_{r,t} \subset \mathbb{R}^{m\times 2^l}$                                          ▷ Sketch matrices
7:     $\{h_t\}_{t\in[T]} : \{0,\dots,2^l\} \to [R]$                                  ▷ hash functions
8: **end members**
9:
10: **public:**
11: **procedure** INIT$(n, d, l, \phi, \epsilon)$
12:     $R \leftarrow \lceil 1/\epsilon \rceil$
13:     $T \leftarrow \Theta(\log(n/\delta))$
14:     $m \leftarrow O(1/\phi^2)$
15:     **for** $i \in [n], r \in [R], t \in [T]$ **do**
16:         $y_{r,t}^i \leftarrow \mathbf{0}$                                                ▷ Sketch vectors
17:     **end for**
18:     **for** $i \in [n], r \in [R], t \in [T]$ **do**
19:         generate $A_{r,t} \in \mathbb{R}^{m\times 2^l}$                              ▷ See [KNW10] for details
20:     **end for**
21:     initialize $h$
22: **end procedure**
23:
24: **procedure** UPDATE$(i \in [n], j \in [d], z \in \mathbb{R})$
25:     $\xi \leftarrow j \cdot 2^l/d$
26:     **for** $t \in [T]$ **do**
27:         $y_{h_t(\xi),t}^i \leftarrow y_{h_t(\xi),t}^i + A_{h_t(\xi),t}\mathbf{i}_{j,z}^l$       ▷ $\mathbf{i}_{j,z}^l$ is define to be the $2^l$-dimensional vector with $j$-th entry of $z$ and others to be 0
28:     **end for**
29: **end procedure**
30:
31: **procedure** SUBTRACT$(i, j, k \in [n])$
32:     **for** $r \in [R], t \in [T]$ **do**
33:         $y_{r,t}^i \leftarrow y_{r,t}^j - y_{r,t}^k$
34:     **end for**
35: **end procedure**
36:
37: **procedure** QUERY$(i \in [n], \xi \in [2^l])$
38:     **for** $t \in [T]$ **do**
39:         $V_{h_t(\xi),t} \leftarrow \mathrm{median}_{\zeta\in[m]}\, y_{r,t,\zeta}^i / \mathrm{median}(|\mathcal{D}_p|)$       ▷ Definition A.5
40:     **end for**
41:     $V \leftarrow \mathrm{median}_{t\in[T]}\, V_{h_t(\xi),t}$
42:     **return** $V$
43: **end procedure**

---

- UPDATE$(i \in [n], j \in [d], z \in \mathbb{R})$*: Update the sketches, running in time* $O(\log(n/\delta))$

- SUBTRACT$(i, j, k \in [n])$*: Subtract the sketches, running in time* $O(\epsilon^{-1}\phi^{-2}\log(n/\delta))$

- QUERY$(i \in [n], \xi \in [2^l])$*: This function will output a $V$ satisfying*

$$(1 - \phi) \cdot F_i(l, \xi) \leq V \leq (1 + \phi) \cdot (F_i(l, \xi) + 5\epsilon\|x_i\|_2^2),$$

*where $F(l, \xi)$ is defined as*

$$F_i(l, \xi) := \sum_{j=\frac{\xi}{2^l} d}^{\frac{\xi+1}{2^l} d - 1} |x_{i,j}|^2,$$

*running in time $O(\phi^{-2} \log(n/\delta))$*

We choose $\phi$ to be constant when we use the above Lemma.