# OpenReview forum: "Fast Distance Oracles for Any Symmetric Norm"
_NeurIPS.cc/2022/Conference — NeurIPS 2022 Accept_

### Official Review · Reviewer_FHL4 · 2022-07-10

**Rating:** 6
**Confidence:** 4
**Soundness:** 3 good
**Presentation:** 3 good
**Contribution:** 3 good

**Summary:**

The paper provides a data structure based on matrix sketches and embeddings for the problem of estimating similarities, and computing distances. Specifically speaking, the proposed idea is to present a distance oracle that is represented by an efficient data structure.

**Questions:**

As mentioned before, my concern is regarding the running time stated in the paper. Can you explain why this is not a problem or suggest a way to reduce this running time to make it much more practical, considering that the similarity problem is highly important in machine learning and other fields as well.

In addition, can coresets fit the role of solving the objective of this paper? specifically dynamic coresets?

**Limitations:**

The proposed data structure takes time that might be astronomical for small error parameters $\varepsilon$.

**Strengths And Weaknesses:**

* Strengths:
   1) The paper proposes a data structure based on a distance oracle for a family of norms, namely, symmetric norms.
   2) The proposed data structure in the paper uses matrix sketches and embeddings to obtain theoretically faster running time than existing techniques.


* Weaknesses:
   1) The proposed data structure takes time that is at least $\Omega\left( \varepsilon^{-9} \right)$. Even for small error accuracies, e.g., $\varepsilon = 0.001$, the running time tends to be astronomical.
   2) The writing of the paper is slightly hard to follow. Please add some paragraphs to ease the transitions, specifically around the algorithms stated in the main paper.

---

### Official Review · Reviewer_p7qi · 2022-07-11

**Rating:** 6
**Confidence:** 3
**Soundness:** 3 good
**Presentation:** 2 fair
**Contribution:** 2 fair

**Summary:**

The authors present a data structure, for a given set of $n$ vectors in a $d$-dimensional metric space and a query vector, that can quickly approximate the distances of the query vector and any vector of the data structure.

The data structure can manage any symmetric norm hence it is not only limited to $\ell_p$ norms. For $\ell_p$ norms the data structure is as efficient as standard oracles, but it can be applied to more norms.

**Questions:**

The authors should add more formal theorems, explanatory or guiding text passages and sketch some of the proofs a little more in detail.

minor errors:
- $\ell \ell$ 46 onward: the sentence is confusing;
- $\ell$ 128: ''Section 4 analyze ...'' should be something like ''\textbf{In} Section 4 \textbf{we} analyze ...''. Same goes for the following sentences in the roadmap.
- $\ell$ 128: \st{Then ,}\textbf{I}t
- $\ell$ 220: \st{When it comes a data update}\textbf{When updating the data with Algorithm 5} ...
- $\ell$ 229: take\st{s}

**Limitations:**

No potentially negative societal impact in sight.

**Strengths And Weaknesses:**

The paper is well written and structured. Giving an intuitive version of the main results and used techniques is also really helpful. The problem the authors tackle is interesting, although it is iterative (i.e. the cases in which their improvements apply are minor).

The result is iterative as mentioned, though it remains interesting. Instead of giving sketches the authors reference knowledge of the appendix. This is even done to main results, which are only formulated in an informal way. Sometimes the reader is left on their own with out any text guiding them through sections, which is irritating.

---

### Official Review · Reviewer_oXwu · 2022-07-12

**Rating:** 7
**Confidence:** 4
**Soundness:** 4 excellent
**Presentation:** 2 fair
**Contribution:** 3 good

**Summary:**

The paper considers the distance oracle problem, where one has a database X of points, and given a query point, one must compute the approximate distances from the query to all points of some subset of X. The authors consider this problem in the very general setting of symmetric norms, norms where the ordering of the coordinates is unimportant, and taking absolute value of the coordinates does not alter the norm. Results are given in terms of the modulus of concentration of the norm.

There are several interesting techniques presented in the paper. The paper builds on the well-known "layered approximation" technique, but optimizes it for the current problem by reducing the number of layers, ensuring linearity, and reduces the decoding time. The overview of these techniques is informative.

**Questions:**

None

**Strengths And Weaknesses:**

Strengths: The problem is interesting, as are the techniques employed and the achieved result.

Weaknesses: Needs a few more rounds of proofreading, and the presentation needs to be improved.

There's too little detail in the body of the paper, and one has to look at the appendix to understand anything about the construction.

The problem statement is strange. Generally, distance oracles are defined on the input set X, to quickly return the distances between any two points in X (i.e. in less than d time). The problem being solved here is a different version of that, where this is a query point not in X, but the authors should not present this as the only (or in fact even the main) meaning of the term distance oracle.

Another issue is the repeating assumption that linearity is somehow central to the functioning of distance oracles. Perhaps it's essential to the current construction, but many oracles do not require linearity. I'm reminded of Indyk's hashing for lp using p-stables, where he takes the median over many computed distances. Oracles for non-vector spaces don't use linearity at all.

Minor comments:
There are some informalities ("cheap", "a bunch") that are better avoided.
p2l52  norm norm
p2l65  n|S| shoud be d|S|
p2l72  "need to design" does not belong in a definition of a structure
p2l83  dst_i isn't defined yet
p3l105  How many 0's are there?
p3 table  for the k-support norm, shouldn't there be a dependence on k?
p4l119  it's a 1+epsilon approximation, not epsilon
p4l133  The problem under consideration (recovering the norm) should be mentioned in the beginning of the paragraph.
p5l11  What's t?
p6  Both algorithms have the same name

---

### Official Review · Reviewer_HT4N · 2022-07-13

**Rating:** 4
**Confidence:** 3
**Soundness:** 3 good
**Presentation:** 1 poor
**Contribution:** 2 fair

**Summary:**

This paper addresses the problem of distance oracles for symmetric norms. In this context, the goal is to preprocess a dataset of $n$ input points $x_1,\ldots,x_n$ in some metric space, into a small-space data structure so that given a query vector $q$ and a subset $S \subseteq [n]$, one can quickly estimate all the distances $d(q, x_i)$. To this goal, the author(s) design a new data structure named to generate sketches and manage them. With this data structure, the author(s) obtain a sublinear-time distance estimation algorithm to solve DO.

**Questions:**

Q1: $||{\cdot}||_\mbox{sym}$ denotes a fixed but arbitrary symmetric norm, correct? I think this should be said somewhere before referring to it as "the" symmetric form.

Q2: On Line 83: What is $\mbox{dst}_i$? It did not seem to have been defined. Same on Line 121. Did I miss the definition?

Q3: On Line 152, it says "$\ell_2$ -heavy-hitters." What does this mean?

Q4: On line 194 and 195, you mention $\epsilon$ and $\delta$ but they are not used in the following sentences. So would it be reasonable to not mention them in those lines to help the exposition?

Q5: On Line 297, what is the order of the $\setminus$ and the $\cup$? Parenthesis would be helpful.


Minor typos

Line 1: Please use $\ldots$ when listing elements and no $\cdots$

Line 8: In the abstract, there should be a "-" in "(1 + $\varepsilon$) distance"

Line 52: norm is repeated twice, "norm norm"

Line 43: Add an extra space after the comma that follows the reference [BYJKS04]

Line 63: Add : at the end of the line to make the rest a complete sentence

Line 72: Do you mean "a symmetric norm" instead of "the symmetric norm" or "a generic, but fixed, symmetric form"?

Line 92: use $\ldots$ instead of $\cdots$ in the definition of $[n]$.

Line 100: Do you mean the $\ell_2$-unit sphere?

Definition 1.2: Consider defining the median of a symmetric norm outside of Def. 1.2 (ideally before). This way, Def. 1.2 is only about mc

Line 106: Add "Maximum modulus of concentration" to the definition to make it consistent with Def. 1.2

Line 116: data is repeated twice

Line 147: Use $\ldots$ instead of $\cdots$ in the definition of $\mathcal{L}(v)$

Line 160: all is repeated twice "all all"


**Limitations:**

The authors do address the limitations of their work in specific sections of the paper. I think they do a very good job in this case.

They don't address potential negative societal impact, but as they point out in the check-list, it is theoretical work and does not seem to have any explicit negative societal impact.

**Strengths And Weaknesses:**

Strengths
I think the data structure they design for linear sketching is interesting in its own right and deserves further study. In their open questions, they point out that the efficiency of their data structure depends on the concentration property of the symmetric norm and whether or not said dependency is necessary.

Weaknesses
It is not clear to me how significant it is from a practical point of view to solve DO problem for a generic symmetric norm. I realize the authors do mention that $\ell_p$ norm are symmetric norms, and that the DO problem is understood in this case. However, in practice, do people use arbitrary symmetric norms for which the DO problem wasn't understood? The theoretical motivation is more clear to me, but not the practical motivation.

The paper was also a bit hard for me to read. $\mbox{dst}_i$ did not seem to be defined anywhere, and even if it is obvious it should be defined for the sake of completeness. There are a couple of distracting typos (see the Questions sections) and overall the exposition is bit cumbersome for me to fully follow the relevance of the results of the paper.

There are also no experiments in the paper. Some of the theoretical guarantees in Section 4 could be moved to the appendices to make space for empirical evidence of the theoretical claims.

---

### Author Response · Authors · 2022-07-30
**Author Response**

We thank the reviewers for their time and their helpful inputs, and we will do our best to address all comments in the final version of our paper.

* **(Reviewer HT4N)** The reviewer wonders *“whether people use arbitrary symmetric norms for which the DO problem wasn't understood?”.*
    There appears to be a misconception here – Our data structure provides faster distance estimation for a large class of well-studied and heavily-used symmetric norms beyond $L_p$ norms, most notably Orlicz norms (modeling sub-gaussian data), Top-$k$ norms and min/sum-mixtures of $L_p$ norms. All these symmetric norms are widely used in practice (see paper’s refs) and we believe that unifying them under a single (optimal) algorithm is valuable, both in theory and practice.

* **(Reviewer HT4N)** Regarding the practicality of our data structure: We wish to emphasize that our paper targets theoretical contributions concerning the distance oracle problem, and the development of novel algorithmic tools with *provable* guarantees, which may inspire practical heuristics for the problem in the future. Our new techniques overcome several barriers in sketching distances, as elaborated in Section 2.

* **(Reviewer oXwu)** The reviewer asserts that *“the problem statement is strange”*, as distance oracles are traditionally defined as data structures that quickly return the distance between any *pair* of points in $X$, whereas the problem being solved here asks to estimate the distance of an *arbitrary* point (possibly outside $X$) to a *subset* of points in $X$.
    This is true, however:
    - In the appendix, we design a data structure for the classical version (see subroutine EstPair()).
    - In $\mathbb{R}^d$, the problem of estimating the normed distance between an arbitrary query point $y$ and a single point $x$ in the dataset $X$, is not interesting, since $\sim d$ time is sufficient and necessary to merely read the query point $y$. By contrast, for a *subset* $S$ of points, this naïve algorithm yields $O(d\cdot |S|)$ query time, whereas our query time is $\sim O(d+|S|)$. We also stress that many learning applications require estimating distances between a single query point and *all* the dataset points, as explained in the introduction.

* **(Reviewer oXwu)** *“Another issue is the repeating assumption that linearity is somehow central to the functioning of distance oracles. Perhaps it's essential to the current construction, but many oracles do not require linearity. I'm reminded of Indyk's hashing for lp using p-stables, where he takes the median over many computed distances. Oracles for non-vector spaces don't use linearity at all.”*

    Indeed, linearity is crucial for
    1. reducing distance-sketching to *norm*-sketching, which is more amenable to the “layer-approximation” and heavy-hitter techniques we develop (see 1st paragraph of Sec 2).
    2. linearity is key to handling distances between an arbitrary (new) query point and a point in $X$.

    Distance estimation in vector-spaces is the most popular setting both in theory and practice, however, we agree that the case of (nonlinear) metric spaces (e.g., graphs) is a very interesting open question. Our tools may be relevant for this case as well, via metric embeddings [Bourgain, Matousek].

* **(Reviewer FHL4)** *“The proposed data structure takes time that is at least $\epsilon^{-9}$. Even for small error accuracies, e.g., $\epsilon= 0.001$, the running time tends to be astronomical… Can you explain why this is not a problem or suggest a way to reduce this running time to make it much more practical, considering that the similarity problem is highly important in machine learning and other fields as well? ”*

    We agree that the polynomial dependence on the accuracy parameter is a drawback for the practicality of our data structure, though we did not attempt to optimize the dependence on $\epsilon$. Moreover, quadratic ($1/\epsilon^2$) dependence is generally inevitable for distance-preserving dimensionality-reduction (e.g., [1]). We wish to emphasize that our paper targets theoretical contributions concerning the distance oracle problem, by introducing new techniques which may inspire practical heuristics for the problem. Our new techniques overcome several barriers in sketching distances, as elaborated in Section 2.

[1] Larsen, Kasper Green, and Jelani Nelson. "The Johnson-Lindenstrauss Lemma Is Optimal for Linear Dimensionality Reduction." *43rd International Colloquium on Automata, Languages, and Programming (ICALP 2016).* Schloss Dagstuhl-Leibniz-Zentrum fuer Informatik, 2016.

---

> ### Comment · Reviewer_HT4N · 2022-08-09
> **Reviewer HT4N's response to rebuttal**
>
> The authors try to explain why their contributions are meaningful and I can appreciate their results better now. I won't change my score due to the presentation issues I mentioned. I noticed that the other reviewers gave it a score of 2 regarding presentation, so overall this makes for a hard paper to read and the motivation could be improved. This is a top-tier conference, and papers published here should be high-quality in terms of the work and the presentation/motivation as well. Moreover, while I acknowledge the author's response stating that theirs is a theoretical contribution that develops "novel algorithmic tools with provable guarantees, which *may* inspire practical heuristics for the problem in the future" (emphasis is mine.) However, in my opinion, the paper still needs to show its value in practice.

---

### Meta-Review · Area_Chair_Y87G · 2022-08-26

**Recommendation:** Accept
**Confidence:** Certain

**Metareview:**

Reviewers found the problem, the results and the techniques (very) interesting. The main concerns were about the practicality of the results (esp. lack of experiments) and presentation (notably various typos). The presentation issues appear to be easily fixable with a careful pass over the paper.  Ultimately, the positives significantly outweighed the negatives.

**Award:**

No

---

### Decision · Program_Chairs · 2022-09-14

Accept